# Systematic analysis of membrane contact sites in *Saccharomyces cerevisiae* uncovers modulators of cellular lipid distribution

Inês Gomes Castro[1†], Shawn P Shortill[2,3†], Samantha Katarzyna Dziurdzik[2,3†], Angela Cadou[4†], Suriakarthiga Ganesan[5], Rosario Valenti[1], Yotam David[1], Michael Davey[2], Carsten Mattes[6], Ffion B Thomas[4], Reut Ester Avraham[1], Hadar Meyer[1], Amir Fadel[1], Emma J Fenech[1], Robert Ernst[6], Vanina Zaremberg[5], Tim P Levine[7*], Christopher Stefan[4*], Elizabeth Conibear[2,3*], Maya Schuldiner[1*]

[1]Department of Molecular Genetics, Weizmann Institute of Science, Rehovot, Israel; [2]Centre for Molecular Medicine and Therapeutics, British Columbia Children's Hospital Research Institute, University of British Columbia, Vancouver, Canada; [3]Department of Medical Genetics, University of British Columbia, Vancouver, Canada; [4]Laboratory for Molecular Cell Biology, University College London, London, United Kingdom; [5]Department of Biological Sciences, University of Calgary, Calgary, Canada; [6]Medical Biochemistry and Molecular Biology, PZMS, Medical Faculty, Saarland University, Homburg, Germany; [7]UCL Institute of Ophthalmology, University College London, London, United Kingdom

**\*For correspondence:**
tim.levine@ucl.ac.uk (TPL);
c.stefan@ucl.ac.uk (CS);
conibear@cmmt.ubc.ca (EC);
maya.schuldiner@weizmann.ac.il (MS)

[†]These authors contributed equally to this work

**Abstract** Actively maintained close appositions between organelle membranes, also known as contact sites, enable the efficient transfer of biomolecules between cellular compartments. Several such sites have been described as well as their tethering machineries. Despite these advances we are still far from a comprehensive understanding of the function and regulation of most contact sites. To systematically characterize contact site proteomes, we established a high-throughput screening approach in *Saccharomyces cerevisiae* based on co-localization imaging. We imaged split fluorescence reporters for six different contact sites, several of which are poorly characterized, on the background of 1165 strains expressing a mCherry-tagged yeast protein that has a cellular punctate distribution (a hallmark of contact sites), under regulation of the strong *TEF2* promoter. By scoring both co-localization events and effects on reporter size and abundance, we discovered over 100 new potential contact site residents and effectors in yeast. Focusing on several of the newly identified residents, we identified three homologs of Vps13 and Atg2 that are residents of multiple contact sites. These proteins share their lipid transport domain, thus expanding this family of lipid transporters. Analysis of another candidate, Ypr097w, which we now call Lec1 (Lipid-droplet Ergosterol Cortex 1), revealed that this previously uncharacterized protein dynamically shifts between lipid droplets and the cell cortex, and plays a role in regulation of ergosterol distribution in the cell. Overall, our analysis expands the universe of contact site residents and effectors and creates a rich database to mine for new functions, tethers, and regulators.

## Editor's evaluation

This manuscript describes an extensive and systematic analysis of membrane contact sites in budding yeast to uncover novel proteins required for tethering organelles and modulation of

membrane contacts. The authors identify over 100 new potential contact site proteins and effectors including proteins associated with the recently discovered plasma membrane-LD (pClip) and Golgi-peroxisome (GoPo) contact sites. Further, the authors identify and characterize novel lipid transport proteins associated with the pClip as well as Lec1, an ER-Lipid droplet contact site associated protein that contains a novel putative lipid-binding domain and may facilitate ergosterol transport between the plasma membrane and lipid droplets.

## Introduction

The hallmark of eukaryotic cells is the presence of organelles as biochemically distinct compartments. However, to coordinate cellular responses effectively, organelles must communicate and work cooperatively. Membrane contact sites play an essential role in this communication by actively tethering two organelles in proximity to each other, thus enabling a direct physical interaction and the exchange of ions, lipids and other small molecules (*Eisenberg-Bord et al., 2016*).

Contact sites are formed between most, if not all, cellular membranes (*Kakimoto et al., 2018*; *Shai et al., 2018*; *Valm et al., 2017*) and are implicated in a growing number of processes from organelle morphology and inheritance, to lipid metabolism and intracellular signaling (*Prinz et al., 2020*). To perform these functions, contact sites harbor a defined membrane composition, enriched with specific lipids and proteins, some of which can form tethers between the opposing membranes, enabling a functional interaction (*Scorrano et al., 2019*). While several contact sites have been observed for decades and we know of multiple tethers and functions for the few well-studied contacts, the formation and function of the majority of contact sites remains poorly understood. One important step towards such an understanding would be to obtain the complete proteome of contact sites.

Several techniques have been developed and adapted to enable the identification of contact site proteins, including proximity labeling assays and systematic high throughput screening approaches (*Cho et al., 2017*; *Shai et al., 2018*; *van Vliet et al., 2017*). Although these approaches generally require previous knowledge of contact site resident proteins, this can be circumvented by the use of synthetic reporters, such as split proteins, targeted to the opposing membranes of two organelles. When in close proximity, these split proteins will interact, leading to either emission of a fluorescent signal (*Cieri et al., 2018*; *Eisenberg-Bord et al., 2016*; *Yang et al., 2018*) or an enzymatic reaction (*Cho et al., 2020*; *Kwak et al., 2020*).

We have previously taken advantage of the split fluorescence protein approach to identify unknown contact sites, and to uncover new tethering proteins for the peroxisome-mitochondria (PerMit) contact site (*Shai et al., 2018*), the vacuole-mitochondria contact site (also called vCLAMP for VaCuoLe And Mitochondria Patch; *Bisinski et al., 2022*) and the nucleus-mitochondria contact (*Eisenberg-Bord et al., 2021*), in *Saccharomyces cerevisiae* (from here on termed yeast). Here, we build on this approach and expand it by using high content screens to systematically analyze effectors and resident proteins of six different contact sites: the PerMit, the Lipid Droplet-Endoplasmic Reticulum (LD-ER) contact (LiDER), the nuclear ER-vacuole junction (NVJ), the peroxisome-vacuole contact (PerVale), the Plasma Membrane (PM, or cortex)-LD contact (pCLIP) and the Golgi-peroxisome contact (GoPo), the latter one not previously characterized. All together, we have identified 158 unique proteins with a potential role in tethering and/or regulation of these six contact sites. While focusing on the pCLIP, we identified three proteins – Fmp27 and Ypr117w (recently renamed Hob1 and Hob2 respectively, for Hobbit homologs 1 and 2), and Csf1, which share homology with the lipid transfer proteins Vps13 and Atg2, suggesting that other members of this family of proteins are localized to contact sites. Additionally, as we explored the LiDER, we identified a protein of unknown function, Ypr097w. We demonstrated that Ypr097w plays a role in regulating the distribution of ergosterol in yeast cells, and suggest the name Lec1 for Lipid-droplet Ergosterol Cortex 1. Collectively, our studies create a powerful new resource for contact site research in the form of tens of new potential contact site proteins that can now be functionally explored. Moreover, they highlight the power of our high content screening approach for discovering novel functions of uncharacterized proteins and for expanding our understanding of contact site biology as well as better grasping how cells actively distribute cellular lipids.

## Results

### Mapping the proteome of multiple contact sites using high throughput microscopy screens uncovers potential residents and regulators

To identify new potential contact site resident proteins and regulators, we used a set of contact site reporters based on a bimolecular fluorescence complementation assay (*Alford et al., 2012*; *Sung and Huh, 2007*) that we and others have previously used to visualize contact sites (*Bisinski et al., 2022*; *Cieri et al., 2018*; *Eisenberg-Bord et al., 2016*; *Eisenberg-Bord et al., 2021*; *Kakimoto et al., 2018*; *Shai et al., 2018*; *Tashiro et al., 2020*). In short, each part of a split-Venus protein is used to tag membrane proteins on opposing organelles. Only in instances where these organelles come into extremely close proximity, as is the case at contact sites, the two parts complement each other to form a full fluorophore. Since any membrane protein with the correct topology (terminus facing the cytosol) can be used for this assay (*Shai et al., 2018*), this approach allows for the visualization of contact sites in the absence of known components. Taking advantage of this we used the previously verified reporters for the PerMit, NVJ, LiDER, PerVale, and pCLIP (*Shai et al., 2018*; *Figure 1—figure supplement 1A*). Additionally, we created and characterized a new reporter for the GoPo (*Figure 1—figure supplement 1B-D*). This contact site is very dynamic, rare, and metabolically regulated (*Figure 1—figure supplement 1B, D*), potentially due to the quick turnaround of Golgi cisternae. The NVJ reporter served as a positive control, as it is one of the more characterized contacts, therefore in an effective screen we would expect to identify many of its known components. The remaining contacts are considerably less well described and presented a new challenge.

To identify potential new contact site proteins, we generated strains expressing each of the above-mentioned reporters and crossed them against a collection of 1165 strains each expressing one mCherry-tagged protein under the strong *TEF2* promoter (*Figure 1A*; *Weill et al., 2018*; *Yofe et al., 2016*). The strains were chosen such that each mCherry-tagged protein is either localized to intracellular puncta – the typical cellular distribution of contact site proteins – or has previously been identified as a contact site protein (*Supplementary file 1*). These strains were mated against each of the 6 reporter strains using an automated mating, sporulation and haploid selection procedure (*Cohen and Schuldiner, 2011*; *Tong and Boone, 2006*) to create six new collections where each haploid strain expresses a split-Venus reporter and one mCherry-tagged protein (*Figure 1A*). Each collection was subsequently imaged by high throughput microscopy and the resulting images were manually analyzed for full or partial co-localization between the contact site reporter and the mCherry-tagged protein (residents), and/or effect of the overexpressed protein on the contact site reporter number and/or brightness (effectors). All strains classified as hits were isolated and re-imaged on a higher resolution spinning disk confocal microscope (Micro 1; see Materials and methods). Only those strains that were still clear hits after this step were included in the final hit list (*Supplementary file 2*). It is important to note that in the case of small punctate organelles (such as LDs, peroxisomes, and Golgi) the resolution of light microscopy does not enable us to differentiate between co-localization to the contact or distribution over the entire organelle. Hence, to reduce false positives, if a known resident protein of such organelles appeared as co-localizing with the reporter in the screen, we did not label it as a 'contact resident' and only considered it a hit in the screen if it was also an 'effector'. It is important to note that since we did not factor in the number or size of each partner organelle on the background of each strain, any overexpressed mCherry-tagged protein that has an effect on organelle size and distribution, may result in changes to contact site number and/or brightness in an indirect manner.

A total of 158 unique proteins were identified across all six contact sites (*Supplementary file 2*, *Figure 1B*). When looking at our positive control, the NVJ (*Figure 1C*), we found 20 proteins that co-localize with and/or affect the contact. Among these were the validated NVJ residents Nvj1, Nvj3, Mdm1 and Lam6 (*Elbaz-Alon et al., 2015*; *Henne et al., 2015*; *Murley et al., 2015*; *Pan et al., 2000*), demonstrating the potential power of our approach. Additionally, overexpression of mCherry-Erg2 reduced the frequency and brightness of the reporter signal, suggesting that modulation of the ergosterol biosynthesis pathway affects the formation of the NVJ. This is particularly interesting, since several NVJ resident proteins play a role in sterol sensing and transport, specifically Lam5, Lam6, and Osh1 (*Gatta et al., 2015*; *Levine and Munro, 2001*; *Murley et al., 2015*), and this contact has also been shown to play a role in the mevalonate pathway by facilitating the assembly of HMG-CoA reductases, and consequently affecting activity of these enzymes, during acute glucose restriction

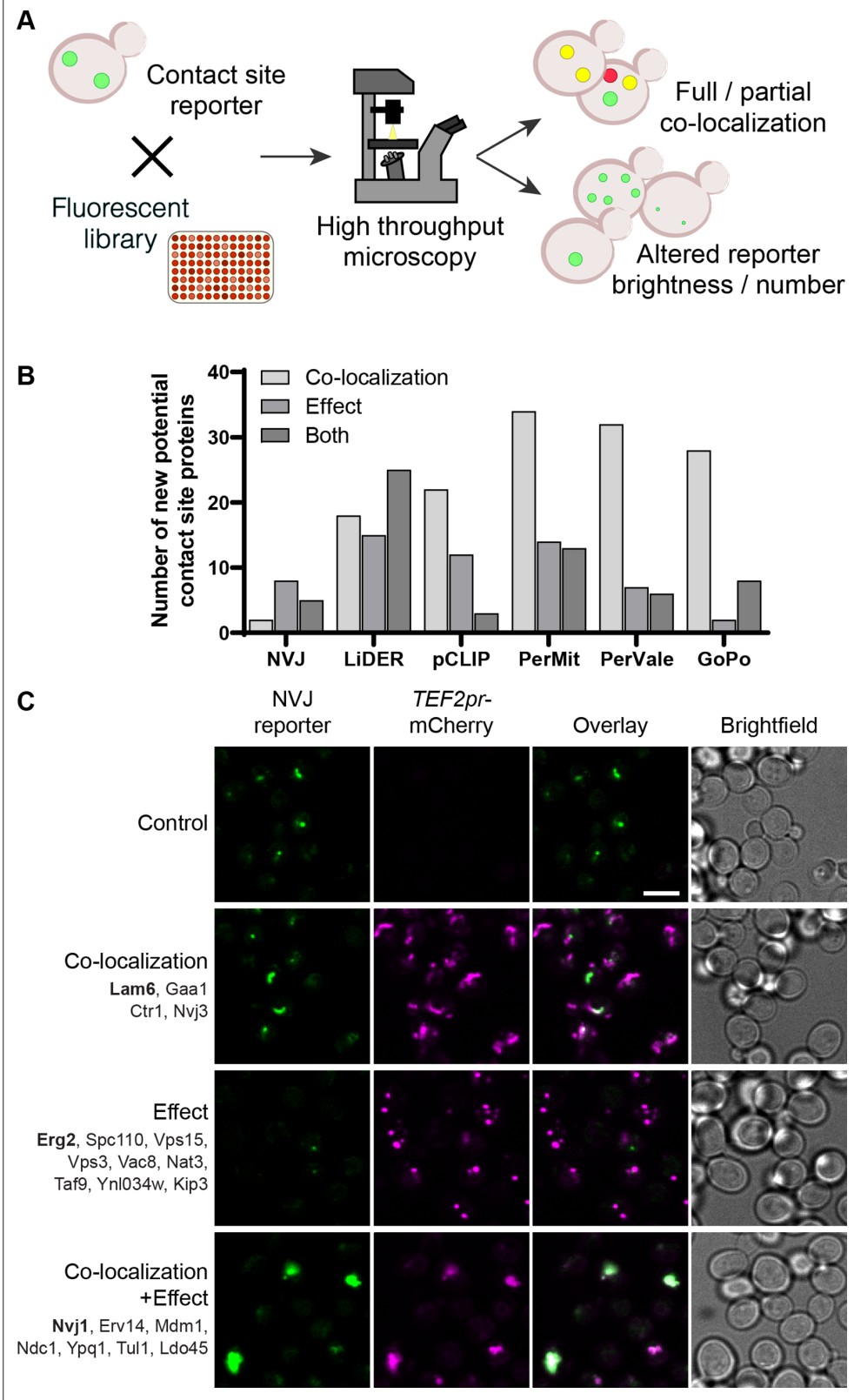

**Figure 1.** Systematic screens uncover new contact site residents and effectors in yeast. (**A**) Schematic representation of the high-throughput screening approach used to identify new contact site residents and regulators. (**B**) Graphical representation of the number of newly suggested contact site residents and effectors (excluding known contact site residents), categorized based on co-localization with and/or effect on contact

*Figure 1 continued on next page*

*Figure 1 continued*

site reporter. (**C**) Summary results of the NVJ contact site screen. A total of 20 proteins were identified and characterized based on their co-localization with and/or effect on the contact site reporter. Individual hits are listed under each category, and example images are from proteins highlighted in bold. Scale bar, 5 µm; Images obtained using Micro 1 (For details of each microscope used see 'Materials and methods' section).

The online version of this article includes the following source data and figure supplement(s) for figure 1:

**Source data 1.** Numerical values used for graph in B.

**Figure supplement 1.** Contact site reporters and GoPo characterization.

**Figure supplement 1—source data 1.** Numerical data used to generate graphs B, D (I) and (II).

---

(*Rogers et al., 2021*). This result supports the ability of our screen to capture not only putative tethers but also effectors.

## The pCLIP contact reveals a superfamily of proteins structurally related to Vps13

One interesting yet uncharacterized contact site in yeast is the pCLIP, between the PM and LDs (*Figure 2—figure supplement 1A*), where we identified 37 proteins that co-localize with and/or affect the contact (*Figure 2A*). To understand how such proteins can function in the contact, we first focused on proteins with known roles in contact sites or with domains that are commonly found in contact site proteins, such as lipid-binding/transfer domains. From these we identified Mdm1, a known contact site protein that localizes to a three-way LD-ER-vacuole contact (*Hariri et al., 2019*). Interestingly, the *Drosophila melanogaster* homolog of Mdm1, Snazarus, also localizes to a triple contact that instead involves LD-ER-PM membranes (*Ugrankar et al., 2019*). Although Mdm1 and Snazarus bind LDs via different domains, and have different specificity for membrane lipids, this new localization for Mdm1 suggests it could also play a role at a yeast LD-ER-PM contact site.

Additionally, we identified a set of 3 proteins that, at the time of our work, were poorly characterized: Fmp27, Ypr117w (closely related paralogs that have now been renamed Hob1 and Hob2, respectively, for <u>Hob</u>bit homologs 1 and 2, due to their homology to the *D. melanogaster* Hobbit protein) and Csf1. We found that these are remote homologs of AsmA, a bacterial relative of Vps13 (*Figure 2—figure supplement 2A*). Advanced modelling indicates that they all share full-length structural homology to Vps13 and Atg2 (*Figure 2B*, *Figure 2—figure supplement 2B*), two well described long, tubular lipid transfer proteins (*Kumar et al., 2018*; *Li et al., 2020*; *Maeda et al., 2019*; *Osawa et al., 2019*; *Valverde et al., 2019*). In yeast, Vps13 has been shown to reside at both the NVJ and the vCLAMP (*Lang et al., 2015*; *Park et al., 2016*). Like Vps13 and Atg2, these proteins are predicted to be long, predominantly hydrophobic, channels, which can potentially transport multiple lipid molecules (*Figure 2B*, *Figure 2—figure supplement 3A*). C-terminally tagged versions of Hob1, Hob2, and Csf1 have recently been shown to localize to ER-PM and ER-mitochondria contact sites (*Neuman et al., 2022*; *Toulmay et al., 2022*). Hob1 and Hob2, and their *Drosophila* homologue Hobbit, target contact sites through N-terminal transmembrane domains that anchor in the ER (*Figure 2—figure supplement 3B*), and C-terminal elements that bind other organelles including the PM (*Neuman et al., 2022*). Csf1 also has a predicted N-terminal ER anchor (*Figure 2—figure supplement 3B*) and has been shown to play a role in lipid homeostasis (*John Peter et al., 2022*). When tagged at the N-terminus, which may interfere with ER targeting, all three proteins co-localize with pCLIP, suggesting that these proteins might play a role at this contact site although we cannot rule out mislocalization due to the tag (*Figure 2C*). Csf1 overexpression appears to decrease the number of detectable pCLIP foci and increase the number of LiDER contacts, indicating possible crosstalk between these sites (*Supplementary file 2*). However, further analysis of this protein is needed to characterize these effects. Moreover, the identification of Hob1, Hob2, and/or Csf1 at several different contacts suggests that these proteins, like yeast Vps13 and human VPS13A, VPS13C, and ATG2A (*Bean et al., 2018*; *Kumar et al., 2018*; *Tamura et al., 2017*; *Tang et al., 2019*; *Yeshaw et al., 2019*), function at multiple localizations.

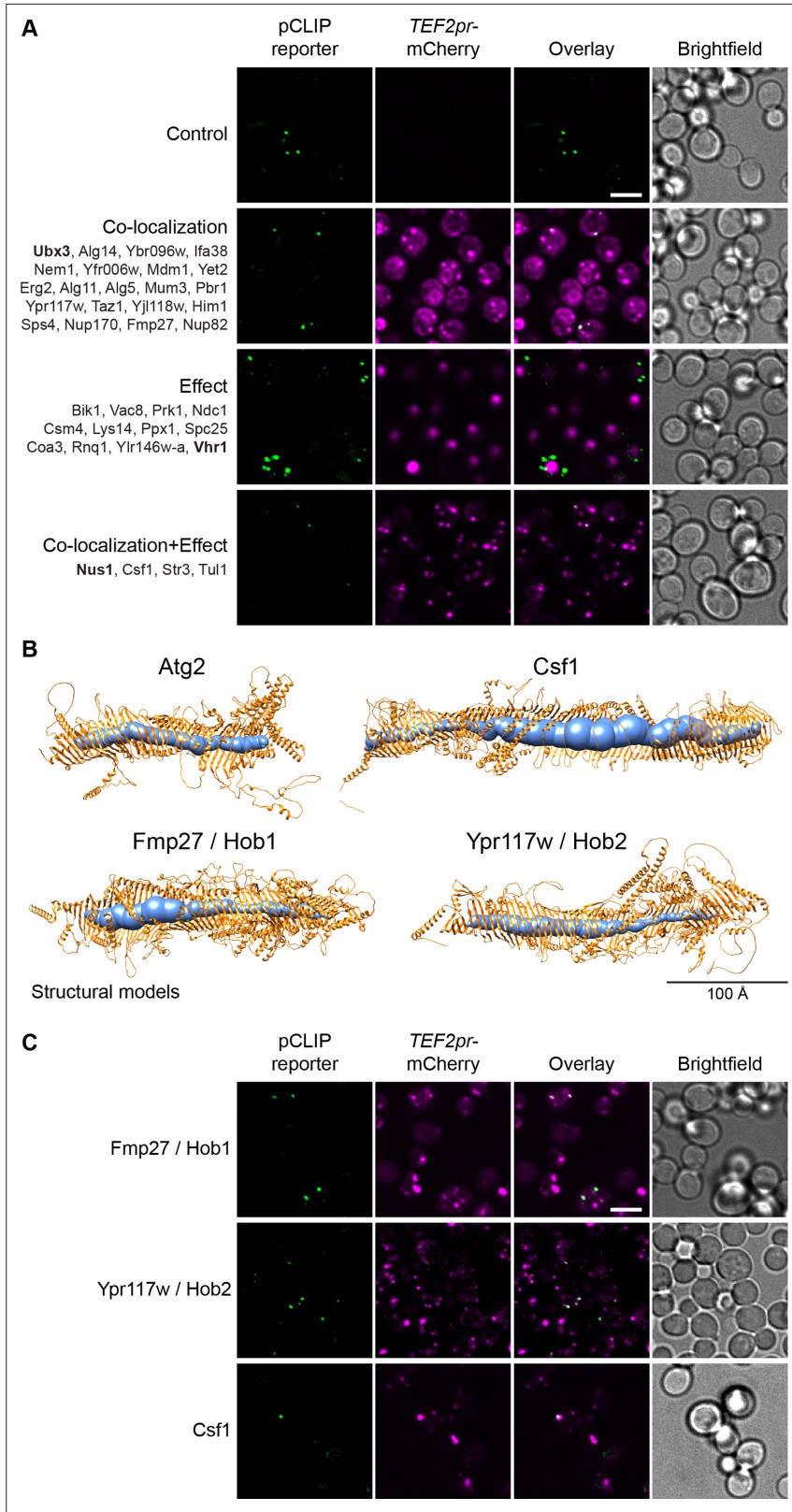

**Figure 2.** The pCLIP screen reveals Vps13 homologs at multiple contacts. (**A**) Summary results of the pCLIP (LD-PM) contact site screen. A total of 37 proteins were identified and characterized based on their co-localization with and/or effect on the contact site reporter. Individual hits are listed under each category, and example images are from proteins highlighted in bold. Scale bar, 5 μm; Images obtained using Micro 1. (**B**) Predicted structures of

*Figure 2 continued on next page*

*Figure 2 continued*

Hob1, Hob2, and Csf1 in comparison to Atg2. Structural predictions were created by the AlphaFold2 consortium for the full-length proteins Hob1/2 and Atg2. For Csf1, four overlapping regions were predicted separately using the AlphaFold2 Colab (*Mirdita et al., 2021*). Internal channel volumes were calculated with MOLE and are depicted in blue. (C) N-terminally mCherry-tagged Hob1, Hob2, and Csf1, under the strong *TEF2* promoter, co-localize with the pCLIP contact site reporter. Scale bar, 5 μm; Images obtained using Micro 1.

The online version of this article includes the following figure supplement(s) for figure 2:

**Figure supplement 1.** pCLIP localization.

**Figure supplement 2.** Vps13 superfamily protein domain structure and comparison.

**Figure supplement 3.** Vps13 superfamily protein channel comparison and membrane topology.

## The LiDER contact screen uncovers a previously uncharacterized protein

One contact that has been little studied is the ER-LD contact site or LiDER. Since LDs remain in close connection to the ER throughout their life cycle in yeast (*Hugenroth and Bohnert, 2020*) this contact is quite prevalent. In our screen the LiDER reporter co-localized with and/or was affected by 59 proteins (*Figure 3A*). Similar to the pCLIP, we focused our analysis on proteins with domains that are common to contact site proteins, such as lipid-binding/transfer domains. As expected, we found the known LiDER protein Mdm1 amongst the hits for this screen. Additionally, we identified Ypr097w, a protein of unknown function that both co-localized with and increased the brightness and frequency of the LiDER reporter in cells (*Figure 3A*, bottom panel). Ypr097w has a Phox homology (PX) domain, which is present in Mdm1 and a large number of proteins conserved across all eukaryotic kingdoms, and which interacts with several anionic lipids, including the phosphoinositide species phosphatidy-linositol 3-phosphate (PI3P) (*Chandra et al., 2019*; *Yu and Lemmon, 2001*). Furthermore, Ypr097w contains a predicted FFAT motif (two phenylalanines, FF, in an acidic tract), a conserved amino acid sequence that enables the interaction of several proteins with the major sperm protein (MSP) domain of VAMP-associated protein (VAP) proteins (Scs2 and Scs22 in yeast) (*Slee and Levine, 2019*). This sequence is present in several contact site proteins, where it enables the formation of contact sites between the ER and other organelles (*Murphy and Levine, 2016*). These characteristics suggest that Ypr097w could function at the LiDER contact.

To confirm the localization of Ypr097w in the absence of a synthetic reporter we re-tagged it at the N-terminus with mCherry under the control of a strong promoter (*Figure 3B*) and imaged its location relative to both the ER and LDs. In these conditions, Ypr097w localized to internal puncta, several of which co-localize with the LD stain, BODIPY, on the interface with the ER (*Figure 3B*), supporting our original observation.

When using the endogenous promoter to tag Ypr097w with GFP at either the N- or C-terminus (both tags support the function of the protein, see Figure 5A later in the article), Ypr097w showed a different distribution than when overexpressed – it was found primarily at buds and bud necks, with reduced localization to LDs (*Figure 3—figure supplement 1*, B). Because endogenous levels were difficult to detect, we also imaged cells expressing GFP-Ypr097w from the moderate constitutive *ADH1* promoter (*Figure 3C*). In this strain, GFP-Ypr097w was more clearly detected and showed a similar distribution pattern to the endogenous promoter, localizing to the bud and bud neck with some increase in the number of internal puncta (*Figure 3C*). Additionally, many cells (nearly 60%) were devoid of a punctate signal and appeared to have a general cytosolic localization. This suggests that the overexpression of Ypr097w either alters cellular physiology or saturates binding sites on the bud/bud neck and causes intracellular accumulation. Interestingly, GFP-Ypr097w localization was also affected by the cell cycle, with a bud/bud neck localization more common in cells with large buds (*Figure 3—figure supplement 1C*).

To further corroborate Ypr097w localization to LDs, we took three complementary approaches. First, we looked at its distribution in cells lacking the LD biogenesis protein seipin by deleting the *FLD1* gene that encodes a subunit of the complex. Seipin mutants form immature LDs that do not detach from the ER. We found that Ypr097w accumulates at LDs in Δ*fld1* mutants (*Figure 3D*), corroborating previously published proteomic analysis of these structures (*Wang et al., 2018*). Second, we performed a genome-wide protein fragment complementation screen based on a split dihydrofolate

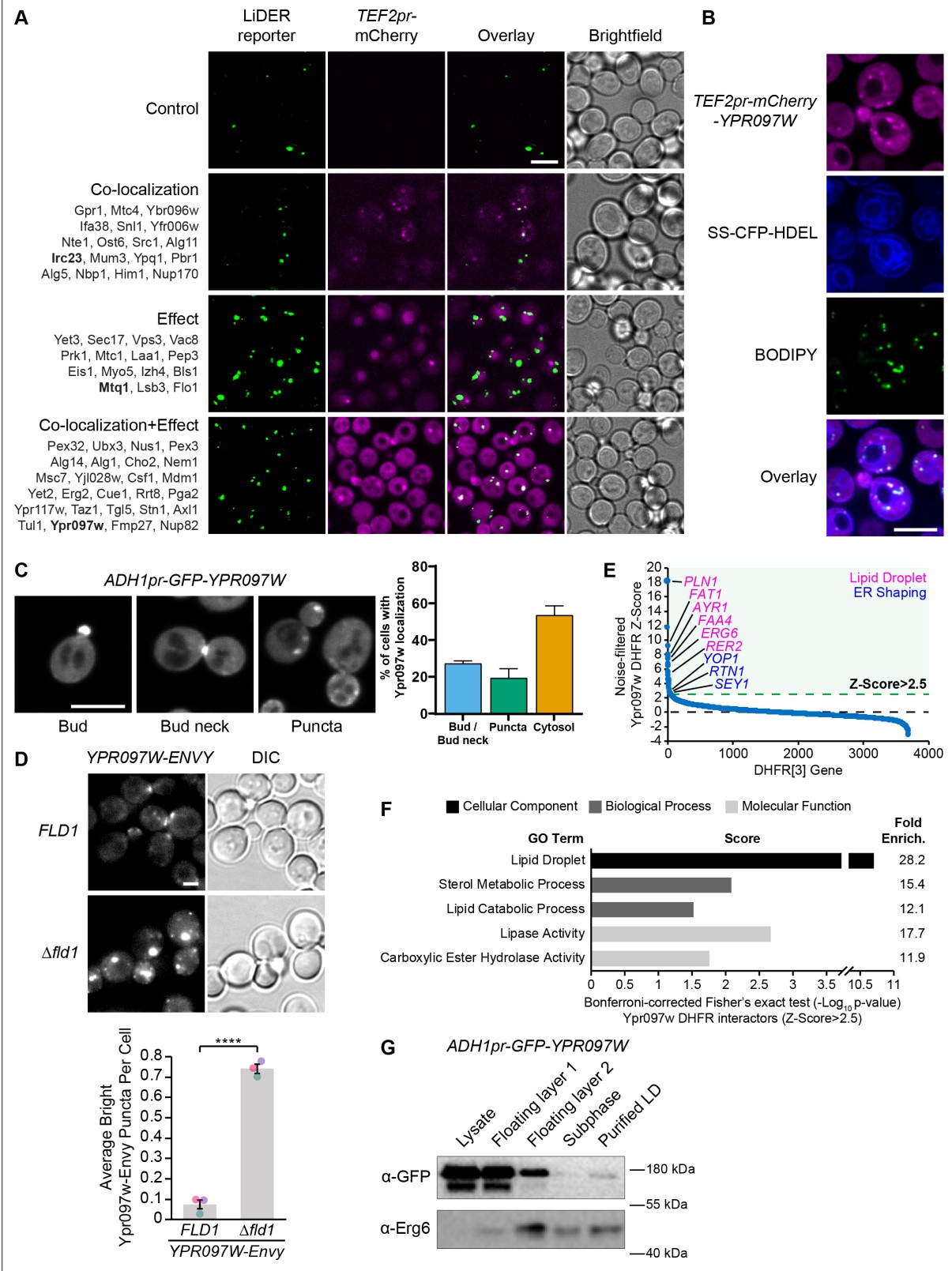

**Figure 3.** The LiDER screen reveals new residents and effectors. (**A**) Summary results of the LiDER contact site screen. A total of 59 proteins were identified and characterized based on their co-localization with and/or effect on the contact site reporter. Individual hits are listed under each category, and example images are from proteins highlighted in bold. Scale bar, 5 μm; Images obtained using Micro 1. (**B**) mCherry-Ypr097w, under control of the *TEF2* promoter, localizes to LDs (BODIPY) in the periphery of the ER (SS-CFP-HDEL). Scale bar, 5 μm; Images obtained using Micro 2. (**C**) Cellular

*Figure 3 continued on next page*

*Figure 3 continued*

localization of *ADH1pr-GFP-YPR097W* during mid-logarithmic growth. Cells were categorized based on Ypr097w localization and quantified as shown in the graph (data are presented as mean ± SEM (n=3)). Scale bar, 5 µm; Images obtained using Micro 1. (**D**) Ypr097w-Envy accumulates at large, bright intracellular puncta in *Δfld1* mutants. Quantification of Ypr097w-Envy puncta per cell. Two tailed equal variance t test; n=3, cells/strain/replicate ≥2448; ****=p < 0.0001. Scale bar, 2 µm. Error bars report SEM. (**E**) Noise-filtered Z-Score distribution of colony area from a DHFR protein fragment complementation assay with endogenously expressed Ypr097w used as a bait. Filtering was used to remove prey strains that exhibited strong signal with a validated cytoplasmic DHFR reporter. An enrichment of proteins with reported LD subcellular localization patterns and proteins with roles in ER shaping was observed in prey strains with a Z-Score of >2.5. (**F**) Functional enrichment analysis of strong Ypr097w DHFR interactors (Z>2.5) using the Gene Ontology (GO) enrichment analysis tool (***Ashburner et al., 2000***; ***Mi et al., 2019***; ***The Gene Ontology Consortium, 2019***). GO terms are presented as the negative base 10 log of the associated p-value from a Bonferroni-corrected Fisher's exact test. Lipid Droplet (GO:0005811); Sterol Metabolic Process (GO:0016125); Lipid Catabolic Process (GO:0016042); Lipase Activity (GO:0016298); Carboxylic Ester Hydrolase Activity (GO:0052689) are significantly enriched ontologies. (**G**) Cells expressing *ADH1pr-GFP-YPR097W* were collected and fractionated by centrifugation to obtain enriched LD fractions. Sequential fractions were run by SDS-page and analyzed by western blot. GFP-Ypr097w is present in the LD fraction. Erg6 was used as a LD marker.

The online version of this article includes the following source data and figure supplement(s) for figure 3:

**Source data 1.** Numerical values used for graph C.

**Source data 2.** Numerical values used for graph D.

**Source data 3.** DHFR source data used for graphs E and F.

**Source data 4.** Original and labeled raw unedited blots for G.

**Figure supplement 1.** Ypr097w is dynamically relocalized in various conditions.

**Figure supplement 1—source data 1.** Numerical values used for graph C.

**Figure supplement 2.** Effect of Ypr097w truncation or mutation on cellular localization.

reductase (DHFR) enzyme (***Tarassov et al., 2008 Supplementary file 3***). This screen, which measures interactions based on colony size after multiple days of growth, showed that the strongest interactors (Z-Score >2.5), which represent proteins in close proximity to Ypr097w, consist primarily of well-characterized LD-localized enzymes (***Figure 3E–F***) and include ER membrane proteins that localize to, and help shape, regions of tubular ER, such as Rtn1, Yop1, and Sey1 (***De Craene et al., 2006***; ***Hu et al., 2009***; ***Voeltz et al., 2006***). In contrast, bud neck-localized proteins were not enriched in this screen. This suggests that under the nutrient conditions experienced in a yeast colony (as compared to cells grown in liquid media and during mid-logarithmic growth), Ypr097w is primarily localized to LDs in the proximity of the ER. Finally, we biochemically purified LDs from cells expressing GFP-Ypr097w under control of the moderate constitutive *ADH1* promoter and found that GFP-Ypr097w is indeed present in purified LD fractions (***Figure 3G***).

These distinct distribution patterns suggest that the localization of Ypr097w is dynamically regulated under different environmental conditions. Indeed, we found that when cells were shifted from nutrient-containing media to phosphate-buffered saline (PBS), both endogenous (***Figure 3—figure supplement 1D***) and *ADH1pr*-expressed (not shown), GFP-Ypr097w rapidly (<2 min) re-localized from bud/bud neck to the cytosol, an effect that was blocked if the PBS was supplemented with glucose. Similarly, longer glucose starvation in synthetic media led to a mostly cytosolic localization of this protein (***Figure 3—figure supplement 1E***). This change in localization could be a response to cytosolic acidification due to glucose depletion, and resulting loss of interaction of Ypr097w with pH sensing membrane lipids (***Shin et al., 2020***). In contrast, in stationary cells, GFP-Ypr097w localized mostly in a punctate pattern (***Figure 3—figure supplement 1F***, left panel), which quickly shifted to a mix of punctate and bud/bud neck upon replenishment with fresh media (***Figure 3—figure supplement 1F***, right panel). The movement of Ypr097w between buds and bud necks to internal puncta (many of which co-localize with LDs) under different growth conditions, suggests that Ypr097w acts as a nutrient sensor and may have a role at LDs primarily in lipid storage conditions.

How is such dynamic localization regulated? We noticed that in the cell lysates (***Figure 3G***) Ypr097w migrated as a doublet, suggesting that Ypr097w is post-translationally modified. In fact, several phosphorylation sites have been identified in Ypr097w in high throughput studies (***Beltrao et al., 2012***; ***Lanz et al., 2021***), including phosphorylation of residue S451 in the region flanking the predicted FFAT motif (***Figure 3—figure supplement 2A***), which has the potential to regulate interaction with VAP proteins (***Kors et al., 2022***; ***Kumagai et al., 2014***; ***Mattia et al., 2020***). Moreover, the punctate

localization of Ypr097w increased in cells lacking both yeast VAP proteins (Δscs2/Δscs22) (*Figure 3—figure supplement 2B*). However, we found that expression of either phosphomimetic (S451E; S451D) or phospho-null (S451A) mutants (*Figure 3—figure supplement 2C*), or a mutant lacking the FFAT motif (Y456A, D458A; data not shown), in a strain lacking endogenous Ypr097w, did not show any clear differences in localization relative to the wild type (WT) Ypr097w control. This suggests that neither phosphorylation of S451 nor the presence of the FFAT motif regulates YPR097w targeting under regular growth conditions. Since the Δscs2/22 effect on Ypr097w localization appears to be indirect, it could instead be mediated by effects of loss of VAP on other processes such as phospholipid synthesis, sterol traffic and PIP consumption (*Loewen and Levine, 2005*; *Stefan et al., 2011*).

Taken together, these results suggest that the localization of Ypr097w is regulated in response to environmental conditions, and that this protein dynamically partitions between the bud/bud neck and LDs. However, what drives this redistribution is not known and will form the basis of future studies.

## Alphafold predicted models of Ypr097w demonstrate a large internal hydrophobic cavity

To understand how Ypr097w is targeted to different cellular locations, we examined its predicted domain architecture. In addition to its PX domain, which harbors the predicted FFAT motif, Ypr097w has an annotated PX-associated domain (PXB) and a domain of unknown function (DUF) 3818 (*Figure 3—figure supplement 2A*; *Mistry et al., 2021*). By testing a series of protein fragments, we found that truncations that perturb either the N or C-termini disrupt protein localization, and we did not identify any domain that was sufficient for localization to either LDs or to the bud/bud neck (*Figure 3—figure supplement 2D*).

Our finding that targeting is affected by truncation of either terminus of Ypr097w, both of which have no known structure, led us to analyze this protein using structural bioinformatics. Structural homology modeling by HHpred (*Gabler et al., 2020*) showed that the unnamed regions bracketing DUF3818 (*Figure 4A*) have similar levels of conservation as the named domains, and that all regions, except the PX domain (which is longer than previously noted) are made up of multiple alpha-helices (data not shown). Together with our truncation results, this suggests that full-length Ypr097w functions as a single unit. However, other than the PX domain, HHpred searches did not find any homology to solved structures or remote homology to other proteins. This implies that Ypr097w forms a previously undocumented fold.

To determine the Ypr097w fold, we used in silico approaches, in particular AlphaFold2 (*Jumper et al., 2021*). AlphaFold2 outputs for Ypr097w and its full-length homologs in both *C. albicans* and *S. pombe* all predict that the different all-helical regions, which appear separate in the one-dimensional map (*Figure 4A*), interact in cis to form a sphere ~7 nm in diameter (*Figure 4B*). A similar sphere is also predicted for a shorter *S. pombe* paralog that has PXB and DUF3818 without a PX domain (*Figure 4—figure supplement 1A*). A key aspect of these structures is the multiple inter-relationships between the different components of the primary structure. In particular, the N-terminal PXB domain folds intimately together with the unnamed C-terminal region (*Figure 4B* and *Figure 4—figure supplement 1A*). The PX domain is oriented to expose its PI3P binding site, which would allow Ypr097w to dock onto membranes (*Figure 4—figure supplement 1B*). Among the five AlphaFold2 models, the PX-negative *S. pombe* homolog is the most confidently predicted (*Figure 4—figure supplement 1C*), implying that the PX domain is unnecessary for the overall fold.

A striking feature of the predicted Ypr097w sphere is that it is hollow with an internal cavity (*Figure 4C–D*). The surface of the cavity is mainly hydrophobic (lipophilic) (*Figure 4C*), a feature that is highly conserved (*Figure 4D*). Multiple tunnels potentially provide access from the cytosol (*Figure 4—figure supplement 1D-E*, arrows). Intimate interactions of N- and C-termini, overall spheroidal shape, a large internal hydrophobic-lined cavity, and tunnels linking the exterior to the interior were also predicted by the trRosetta server (data not shown; *Yang et al., 2020*). Among the five proteins modelled by AlphaFold2, *S. cerevisiae* Ypr097w is unique in having three disordered loop inserts in its PX domain (*Figure 4A*). Part of the first loop was modelled as dipping into the cavity (*Figure 4—figure supplement 1D*, modelled in white). However, this loop is only found in the budding yeast protein and in that model it is the only part of the cavity predicted with low confidence, at a level AlphaFold identifies as uninformative (*Figure 4—figure supplement 1E*). Therefore, the position of the loop was assumed to be largely external, a position that was supported by trRosetta. We calculated the internal

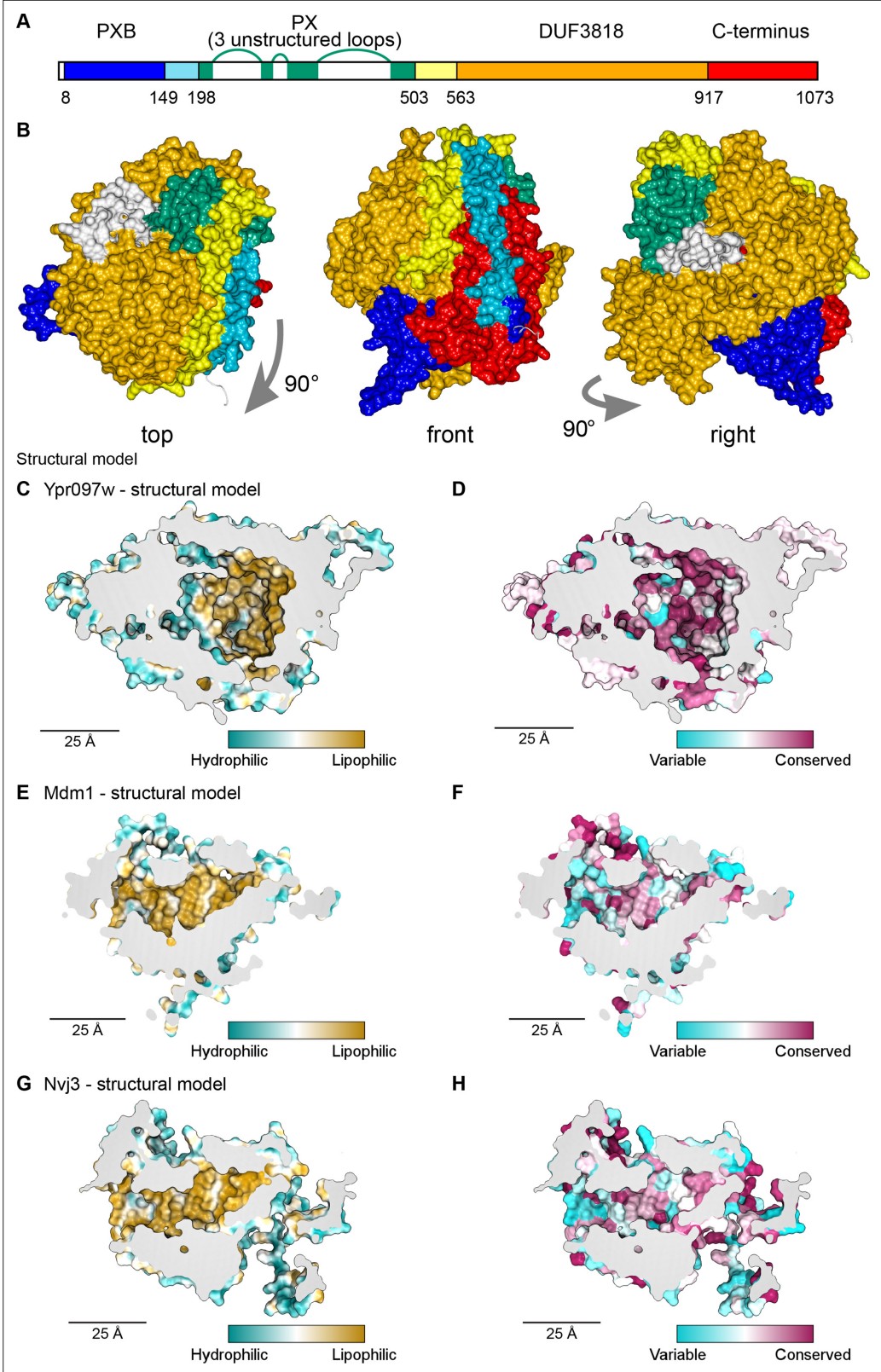

**Figure 4.** Structural bioinformatics predicts that Ypr097w forms a large spherical hydrophobic cavity. (**A**) Domain map of Ypr097w based results from HHpred. The PX domain (shown in green) is extended more towards the N-terminus than shown previously through identifying residues 203–212 as its first strand. The extreme C-terminus (red) is as helical and as conserved as the preceding DUF3818 domain (orange). The PX domain contains three

*Figure 4 continued on next page*

*Figure 4 continued*

loops without strong structural predictions (white). (**B**) AlphaFold2 structure prediction for Ypr097w, colored by domain as in (**A**), omitting loops outside the protein core. Three views from different angles. Domains are intimately associated with each other, in particular the PXB (blue) and extreme C-terminus (red). (**C–D**) Representations of the cavity of Ypr097w predicted by AlphaFold2, with the lining colored by either hydrophobicity (**C**) or conservation (**D**). (**E–F**) Representations of the cavity of Mdm1 predicted by AlphaFold2, coloring as for C/D, respectively. (**G–H**) Representations of the cavity of Nvj3 predicted by AlphaFold2, coloring as for C/D, respectively.

The online version of this article includes the following source data and figure supplement(s) for figure 4:

**Figure supplement 1.** Details of AlphaFold2 structural predictions of Ypr097w in various species.

**Figure supplement 2.** Mutagenesis analysis of Ypr097w hydrophobic cavity.

**Figure supplement 2—source data 1.** Original and labeled raw unedited blots for B.

---

cavity without the loop and found that it has a volume of 29 nm$^3$ (***Figure 4—figure supplement 1C***), equivalent to a sphere diameter of 3.8 nm. Mutating the hydrophobic residues in the binding pocket altered the capacity of Ypr097w to localize to LDs, as well as its functionality (***Figure 4—figure supplement 2A-D***), suggesting the importance of the hydrophobic pocket for structural stability. While this α-helical structure is unique, the cavity resembles those in the family of large lipid transfer proteins (LTPs) related to vitellogenin, including apolipoprotein B (***Smolenaars et al., 2007***). In these proteins, similar sized cavities lined by β-sheets function in passing lipids from membrane sources to diverse destinations.

Given the remarkable structure of Ypr097w, and the now readily available structural models predicted by AlphaFold2, we decided to use a candidate approach to look for additional spherical domains with large lipophilic cavities among known contact site proteins in yeast, including our >100 hits. We noted that the contact site protein Mdm1, similarly to Ypr097w, has a PX domain and two associated domains of unknown structure – the PXA and the nexin-C domains – which are found only in Mdm1 and its homologs, including Nvj3 which like Mdm1 also has a PXA domain and nexin-C domain (identified here using HHpred) at its N and C-termini. Strikingly, AlphaFold2 analysis of these proteins revealed that the PXA and nexin-C domains in Mdm1 and Nvj3 fold together to form a large spherical domain with an interior hydrophobic surface (***Figure 4E and G***) that is highly conserved (***Figure 4F and H***). While the PXA/nexin-C domains share no detectable sequence homology to Ypr097w, the structural similarities suggest that these proteins may share a function that we hypothesize may be in lipid storage or transfer.

## Ypr097w affects the distribution of ergosterol in cellular membranes

Since Ypr097w is localized to LDs, we assayed whether it has any effect on this organelle and its ability to store lipids. To test this, we first looked at the number of LDs in WT, Δ*ypr097w* and overexpressed *YPR097W* (under control of the *TDH3pr*) cells using the LD marker BODIPY. However, we observed no clear changes in the number of LDs (***Figure 5—figure supplement 1A***) or LD morphology (***Figure 5—figure supplement 1B***) in the tested conditions.

To test for changes in the total levels of storage lipids, we performed thin layer chromatography (TLC) of WT, Δ*ypr097w* and *TDH3pr-YPR097W* during mid-logarithmic growth, in stationary cultures (24 hr and 48 hr), and following growth resumption during new media replenishment (***Figure 5—figure supplement 1C***). When compared to control cells, no clear changes in the levels of diacylglycerol (DAG), triacylglycerol (TAG), ergosterol (Erg), or sterol esters (SE) were observed in any of the conditions tested. Additionally, lipidomic analysis of the overexpression strain relative to a control strain also did not uncover any major global lipid differences (***Figure 5—figure supplement 1D***, ***Supplementary file 4***). Overall, this suggests that Ypr097w does not affect the global cellular levels of tested lipids.

While the overall lipid levels were not altered, we wondered if their distribution may be. To test this, we assayed the sensitivity of these strains to the drug Amphotericin B (AmB), which has been shown to bind accessible ergosterol at the PM (***Gray et al., 2012***), and could therefore reveal changes in cellular levels and distribution of free ergosterol. Surprisingly, despite the lack of changes in overall ergosterol or SE levels observed by TLC and lipidomics, loss of Ypr097w sensitized cells to AmB (***Figure 5A***).

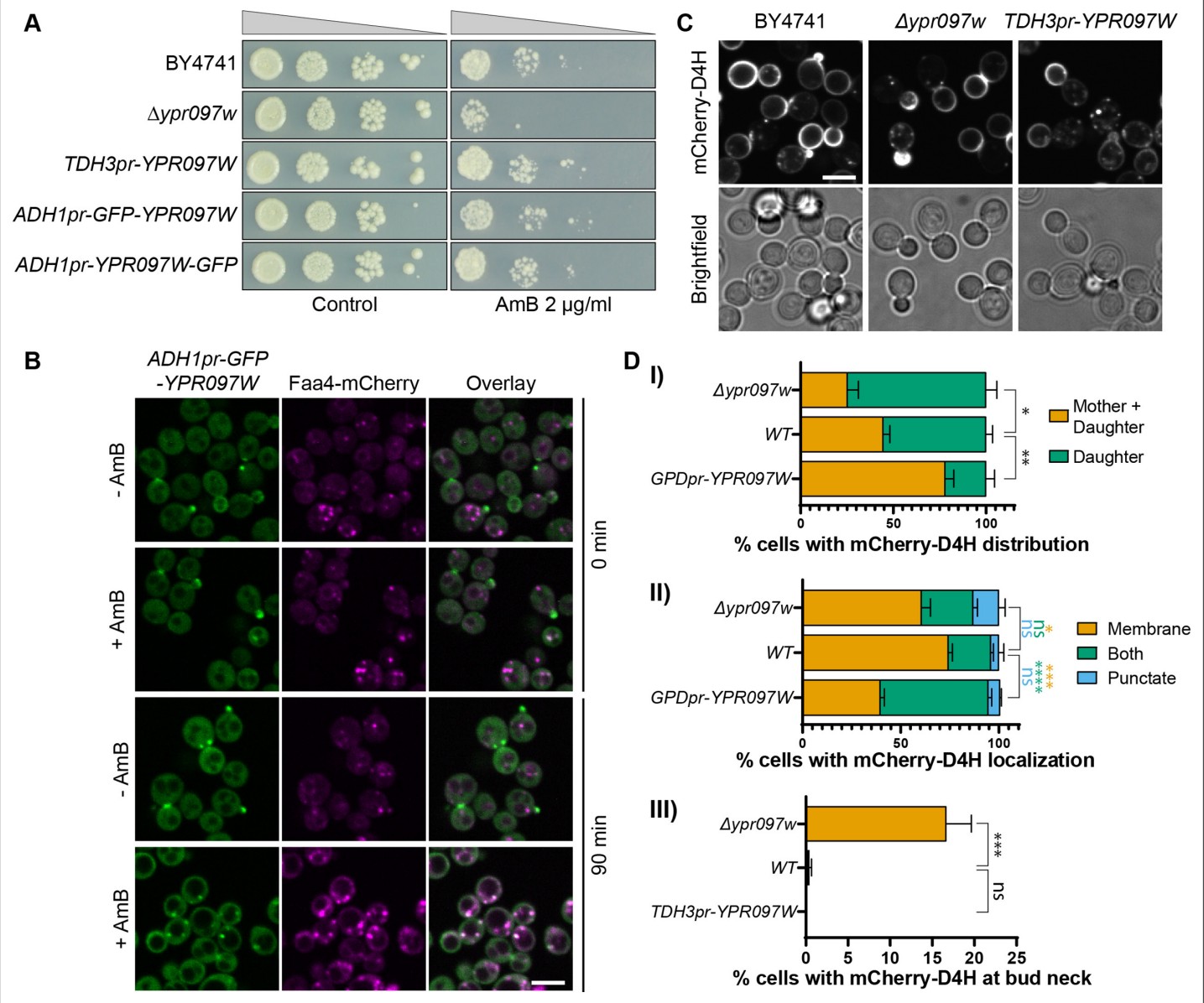

**Figure 5.** Ypr097w affects the cellular distribution of ergosterol. (**A**) Serial dilution growth assay in the presence/absence of Amphotericin B (AmB) (2 µg/ml), after 3 days of growth. (**B**) Localization of *ADH1pr-GFP-YPR097W* in control (DMSO) and AmB treated cells (2 µg/ml), at 0 min and after 90 min continuous drug exposure. Scale bar, 5 µm; Images obtained using Micro 2. (**C**) Cellular localization of the free ergosterol marker mCherry-D4H in WT, *Δypr097w* and overexpression *TDH3pr-YPR097W* cells. Cells were categorized based on marker distribution (mother + daughter cells or daughter) and localization (membrane, puncta or both), as well as accumulation at the bud neck. Scale bar, 5 µm; Images obtained using Micro 2. (**D**) Quantification of three independent experiments relative to (**C**). Data are presented as mean ± SEM (n=3). Analysis performed using ordinary one-way ANOVA with Dunnett's multiple comparison test. ns. not significant, * p≤0.05, ** p≤0.01, *** p≤0.001, **** p≤0.0001.

The online version of this article includes the following source data and figure supplement(s) for figure 5:

**Source data 1.** Numerical values used for graphs D (I), (II), and (III).

**Figure supplement 1.** Ypr097w does not dramatically impact the total level of cellular lipids.

**Figure supplement 1—source data 1.** Numerical data used for graph A.

**Figure supplement 1—source data 2.** Original and labeled raw unedited blots for C.

**Figure supplement 2.** Ypr097w localization is affected by changes in free ergosterol.

**Figure supplement 2—source data 1.** Numerical data used for graph B.

**Figure supplement 3.** Changes in phosphatidylserine (PS) affect free ergosterol and Ypr097w distribution.

Since we have previously seen that Ypr097w is highly dynamic in its localization, we wondered if the cellular trigger for dynamics could be the free ergosterol levels on the PM. To assay this, we tracked the localization of GFP-Ypr097w in cells treated with AmB for 90 minutes and found that indeed, this resulted in a redistribution of Ypr097w (*Figure 5B*). Under these conditions, Ypr097w almost exclusively localized to LDs (as marked by Faa4-mCherry). To complement this approach, we tested the effect of fluconazole, a known inhibitor of Erg11 and ergosterol synthesis, on the localization of GFP-Ypr097w (*Figure 5—figure supplement 2A-B*). To directly track ergosterol distribution in these cells, we constitutively expressed an optimized fragment of the bacterial toxin perfringolysin O, D4H, bound to a mCherry fluorophore. This sensor has been shown to bind free sterols on membranes with a concentration of at least 20 mol% sterols (*Johnson et al., 2012*; *Maekawa and Fairn, 2015*) and binds sterols at the PM in both fission and budding yeast (*Kishimoto et al., 2021*; *Marek et al., 2020*). As expected, fluconazole treatment strongly reduced the level of cellular ergosterol, as shown by the loss of binding of mCherry-D4H to the plasma membrane (cytosolic signal) (*Figure 5—figure supplement 2A* lower right panels). In these conditions, where internal ergosterol is also depleted, we actually see an increase in the number of cells with bud/ bud neck localization for GFP-Ypr097w (*Figure 5—figure supplement 2A-B*). The GFP-Ypr097w signal in these cells generally appeared brighter than in untreated cells, suggesting that loss of internal ergosterol potentiates Ypr097w accumulation at the bud / bud neck.

These results suggest that Ypr097w reacts to changes in ergosterol distribution and in turn, may directly or indirectly affect the distribution of ergosterol in the cell, the levels of free ergosterol, or its capacity to be mobilized.

To understand if Ypr097w affects ergosterol distribution in cells, we looked at the localization of mCherry-D4H in WT, Δ*ypr097w* and overexpressed *YPR097W* (under control of the *TDH3pr*) cells. In WT cells, this sterol reporter is mostly localized to the PM, with an enhanced signal at the daughter cell (*Figure 5C*). Additionally, small punctate structures can also be observed in some cells, generally in proximity to the PM (for examples of phenotypes, see *Figure 5—figure supplement 2C*). When *YPR097W* was either deleted or overexpressed, we observed striking effects on the distribution of the mCherry-D4H sterol reporter. In the absence of Ypr097w, a stronger localization to daughter cells was observed, with the additional accumulation of signal at the bud neck, a phenotype that is mostly absent from WT cells in the conditions tested (*Figure 5C, D*, (III)). This shift in localization is quite intriguing since Ypr097w itself is localized at the bud and bud neck in many cells. This could suggest that the presence of Ypr097w either shifts the distribution of ergosterol on the membrane, or protects it from interacting with mCherry-D4H. In cells where *YPR097W* is strongly overexpressed under the *TDH3* promoter, the distribution of mCherry-D4H was less polarized, with many cells displaying a similar signal between mother and daughter cells. Additionally, these cells showed a much stronger punctate or mixed pattern than both WT and Δ*ypr097w*, suggesting that overexpression of this protein affects the distribution of accessible ergosterol.

Our assays cannot differentiate whether the effect of Ypr097w on ergosterol distribution is direct or indirect. It has recently been shown in mammalian cells that the PM distribution of cholesterol can be significantly affected by the levels of phosphatidylserine (PS) (*Trinh et al., 2020*; *Li et al., 2021*; *Thomas et al., 2022*). Indeed, when we actively decrease the levels of PS on the PM by deleting both Osh6 and Osh7 (*Figure 5—figure supplement 3A*; *Maeda et al., 2013*) we do see changes in the distribution of both mCherry-D4H and Ypr097w. Hence, to see if the ergosterol changes that we observe in Ypr097w mutants are simply a side effect of changing PS levels, we examined the cellular distribution of PS in cells lacking and overexpressing *YPR097W* (*Figure 5—figure supplement 3B-C*). Taking advantage of the PS probe GFP-Lact-C2, we showed that deletion or overexpression of *YPR097W* does not affect the distribution of PS at the plasma membrane (*Figure 5—figure supplement 3B-C*). In addition, cells overexpressing *YPR097W* show no significant differences in the total amount of PS (*Figure 5—figure supplement 3D*), even compared at the level of PS species (*Figure 5—figure supplement 3E*), acyl chain saturation (*Figure 5—figure supplement 3F*), or acyl chain length (*Figure 5—figure supplement 3G*).

Thus, while Ypr097w does not appear to regulate total ergosterol and sterol ester levels (*Figure 5—figure supplement 1C-D*), it appears to play a significant role in ergosterol accessibility, membrane bilayer organization, or localization in the cell. While we cannot still rule out that this is indirect, it is clearly not downstream of PS levels. Due to its distribution and effect on ergosterol we recommend the name Lec1 (Lipid-droplet Ergosterol Cortex 1) for Ypr097w.

## Discussion

There has been a growing interest in characterizing membrane contact sites and understanding their functions. This has been fueled by the development of new imaging and biochemical techniques and by the realization that contact site malfunction underlies multiple diseases (*Castro et al., 2018*; *Herker et al., 2021*; *Scorrano et al., 2019*; *Zung and Schuldiner, 2020*). However, identifying new contact sites and the proteins involved in their tethering and function still represents an important challenge in the field. Here, by taking advantage of split-Venus contact site reporters and high throughput techniques, we have significantly extended the pool of potential contact site resident proteins and effectors for six different contact sites in yeast. Of note is that our screens were far from being saturating since we only screened ~1/3 of yeast proteins, albeit those with the highest chance of being direct residents and effectors. Even with only a subset of yeast proteins, our screens identified a large number of interesting candidates. Screening the additional proteins in the future may uncover more distal regulators such as enzymes that post-translationally modify contact site proteins or dedicated transcription factors. As a clear example of the effectiveness of this technique, we have detected previously known contact site proteins at the NVJ (Nvj1, Mdm1, Nvj3, and Lam6), at the LiDER (Mdm1), and at the PerMit (Pex34 and Fzo1). We anticipate that the newly reported residents and regulators will help drive further work on well-known contacts as well as enable the characterization of the three least studied contact sites – PerVale, pCLIP, and GoPo.

We used this screening approach to identify potential roles for poorly characterized or uncharacterized proteins and report t new family of Vps13 related proteins: Hob1 (formerly Fmp27), Hob2 (formerly Ypr117w), and Csf1. Similar to Vps13, these proteins may localize to multiple contact sites. Thus, these newly discovered members of the Vps13/ATG2 superfamily, recently identified by others (*Neuman et al., 2022*; *Toulmay et al., 2022*), could potentially play similar roles in lipid transport between membranes. This function appears to be conserved in *Drosophila*, where a single homologue of the Hob1 and Hob2 proteins was initially described to play a role in intracellular trafficking (*Neuman and Bashirullah, 2018*). Furthermore, Csf1 was recently shown to function in lipid homeostasis under cold stress and in cells with rewired glycerophospholipid synthesis pathways (*John Peter et al., 2022*), and has been implicated in the transport of phosphatidylethanolamine from other organelles to the ER (*Toulmay et al., 2022*). It is interesting to speculate that like Vps13, this set of related proteins may similarly function in glycerophospholipid transport at contact sites (*Kumar et al., 2018*; *Li et al., 2020*). Additionally, these proteins have homologs in mammalian cells – BLTP2 (for Bridge-like Lipid Transfer Protein family member 2) (also known as KIAA0100/BCOX1) for Hob1/2, and BLTP1 (also known as KIAA1109) for Csf1. Both of these have been implicated in human disease (*Gueneau et al., 2018*; *Kane et al., 2019*; *Liu et al., 2014*; *Song et al., 2006*).

While exploring the LiDER contact, we identified Ypr097w/Lec1, a previously uncharacterized protein. This protein stood out as it has a PX domain for binding lipid headgroups, as well as one of the strongest FFAT motifs in yeast (*Loewen et al., 2003*; *Slee and Levine, 2019*). Despite a clear dependence on yeast VAP proteins for its cellular distribution, we were unable to establish a role for the FFAT motif in directing Lec1 localization under the conditions that we tested. However, we determined that Lec1 appears to act as a sensor of nutrient status, changing its cellular localization under multiple metabolic conditions, highlighting the ability of our screen to identify even transient contact site residents.

A recent *ab initio* predicted structure of Ypr097w (AlphaFold2; *Jumper et al., 2021*) shows Lec1 is composed primarily of alpha helices that fold into a globular structure, with the PX domain on one side. The N- and C-terminal regions of the protein are highly interwoven, which may explain how N- and C-terminal truncations disrupt its localization. Strikingly, the predicted structure of this protein shows a large internal hydrophobic cavity, with multiple channels opening to the cytosol. The hydrophobicity of the residues lining the pocket is highly conserved and functionally important. This tantalizing structure suggests that Lec1 may represent a novel fold that binds or transports lipids.

There is a precedent for an all-helical protein that forms a lipid binding cavity. Such a cavity has been described for the insect allergen repeat domain, which contains two repeats of a domain of 5 helices, forming two hemispheres with one lipid inside (*Mueller et al., 2013*). The Lec1 cavity is uniquely large for a globular protein, at 29 nm$^3$ almost 10-fold larger than any known cavity with a predominantly hydrophobic surface (*Chwastyk et al., 2020*). Since Lec1 is associated with membranes, LDs and also with lipid modifying enzymes, we consider it possible that it encloses hydrophobic molecules such

as lipids or intermediates of lipid synthesis. The cavity volume is over an order of magnitude bigger than the volume of a single lipid molecule. Hence, we hypothesize that Lec1 could potentially accommodate a mixture of many different neutral lipids (e.g. triolein 1.62 nm$^3$) and amphipathic lipids (e.g. DOPC 1.3 nm$^3$). Interestingly, we found two additional proteins that share this predicted form (using AlphaFold2) - Mdm1 and Nvj3, which fold together to form large spherical domains with an interior hydrophobic surface (*Figure 4E and G*) that is highly conserved (*Figure 4F and H*). While the PXA/ nexin-C structure is similar to Lec1, they share no detectable sequence homology suggesting that other proteins with such a fold may exist. A more complete survey of predicted structures may reveal yet more proteins with large hydrophobic cavities that may function in lipid binding or transport. While this manuscript was under revision, an additional group characterized the structure of Lec1 based on the AlphaFold2 prediction, and noted its similarities to Mdm1 and Nvj3. The authors arrived at the similar conclusion that Lec1 might play a role as a LTP (*Paul et al., 2022*).

We found that Δ*lec1* cells are sensitive to AmB, an antifungal drug that interacts with free ergosterol at the PM and is suggested to induce membrane permeabilization and ergosterol removal (*Anderson et al., 2014*; *Gray et al., 2012*). Sensitivity to AmB has also been observed for other contact site proteins, such as Lam1 (Ysp1), Lam2 (Ysp2) and Lam3 (Sip3), which localize to ER-PM contact sites and are involved in the retrograde transport of ergosterol from the PM to the ER (*Gatta et al., 2015*). By taking advantage of an ergosterol-binding fluorescent probe, we show that changes in the levels of Lec1 affect the localization of free ergosterol in the cell, affecting its distribution between mother and daughter cells as well as internal reservoirs. The stronger accumulation of accessible ergosterol at the bud and bud neck of cells lacking Lec1 could explain how these mutants are more sensitive to AmB, as higher levels of accessible ergosterol would make budding cells more sensitive to this drug. Together with the lack of changes in overall ergosterol levels in these cells, our results suggest that Lec1 plays a role in the transbilayer organization or mobilization of ergosterol between membranes in yeast. Consistent with this, upon addition of AmB that binds accessible ergosterol, Lec1 shifts from buds and bud necks to LDs where it may either sequester or mobilize sterols. Intriguingly, a complementary approach using fluconazole to decrease the overall cellular levels of ergosterol led to an increase in Lec1 at the bud / bud neck instead of internal structures such as LDs. This is possibly due to the fact that fluconazole also affects internal ergosterol levels and not just free ergosterol on the PM. However, as both of these drugs cause pleotropic effects, these results might also be an indirect response. Further investigation will be necessary to identify the mechanisms that regulate Lec1 localization in cells and to understand how this affects ergosterol distribution.

More generally our results show that despite the large volume of work on contact sites, multiple contact residents and regulators still await exploration. We anticipate that our work will provide a rich ground for further exploration of the contact site machinery.

## Materials and methods

### *S. cerevisiae* strains and plasmids

*S. cerevisiae* strains, plasmids and primers used in this study are described in *Supplementary files 5-7*. Yeast strains were constructed from the laboratory strain BY4741 (*Brachmann et al., 1998*). Cells were genetically manipulated using the lithium acetate, polyethylene glycol (PEG) and single-stranded DNA (ssDNA) method for transformation (*Gietz and Woods, 2006*). Strains created using organelle markers expressed from plasmids were made fresh for imaging. Primers for genetic manipulations and validation were designed using Primers-4-Yeast (*Yofe and Schuldiner, 2014*). Plasmids expressing Ypr097w-GFPEnvy truncations were made by homologous recombination in yeast by co-transforming linearized plasmids with generated PCR products. The plasmids were recovered in *Escherichia coli* and sequenced. Plasmid MS1060 expressing GFP-Ypr097w WT was generated by isolation of the gene including 3' and 5' UTR by digestion of plasmid MD440 with ApaI and SacI, and cloning to plasmid MS889. Addition of N-terminal yeGFP was performed by restriction free cloning using primers 1060_ GFP_F and 1060_GFP_R, and plasmid MS246. Plasmids expressing Ypr097w phosphomutants were generated by overlap PCR and cloning using XbaI and NsiI restriction enzymes. In short, a fragment from plasmid MS1060 was amplified using the forward primer "YPR_seq2F" and three different reverse primers: 1065_S451E_NsiI_R, 1066_S451D_NsiI_R and 1067_S451A_NsiI_R, one for each mutation, respectively. PCR products and the original plasmid

MS1060 were digested using XbaI and NsiI, and ligated to generate the three mutant plasmids. C-terminally GFP-tagged versions of Ypr097w with or without multiple mutations without disturbing the upstream region that includes the gene SYT1 were created by inserting a construct at the endogenous YPR097W locus, using *C. albicans* URA3 as selection marker. An integration construct was constructed replacing the D4H open-reading frame that follows URA3 in pMS1093 by 500 bp 3' of YPR097W (pTL699). Then, YPR097w was inserted before URA3 with two overlapping fragments synthesized by Twist Bioscience (San Franscisco, USA), including 245 bp 5' UTR, the YPR097w open reading frame with 13 mutations to add 10 and remove 3 restriction sites, the synthetic 39 bp terminator defined as $T_{Guo1}$ (pTL700) (*Curran et al., 2015*). Super-folder GFP was added between open-reading frame and terminator using a further synthesized fragment (pTL701). 45 residues were changed either all to alanines or all to serines using fragments (19 residues in N-terminal half: F42, L79, V82, I87, F90, L92, F103, W104, F111, F112, F115, F120, L139, V142, L145, L146, W250, F255, F274; 26 residues in the C-terminal half: F553, L567, V569, I571, F587, W590, Y594, L619, F648, F663, W807, I819, I823, F831, F843, F846, F885, F888, F901, F904, W907, I908, I911, F1005, L1007, F1024). 19 A/S and 26 A/S plasmids were made by swapping segments between wild-type and 45 A/S plasmids. A null allele was created with 3 nonsense mutations (following residues 22, 128 and 282) introduced by adding/removing residues at individual restriction sites in pTL700 using the Klenow fragment of DNA polymerase; similarly residues 691–1073 were deleted cutting at two sites and re-ligating with Klenow (pTL708).

All insertants were checked by diagnostic digest of a colony PCR. All oligonucleotides used in this study to generate PCR products for cloning are listed in *Supplementary file 7*.

## Growth conditions and microscopy

Yeast cells were cultured overnight in synthetic minimal media (SD) (0.67% [w/v] yeast nitrogen base with ammonium sulfate, 2% [w/v] glucose, amino acid supplements) at 30 °C, unless stated otherwise. Amino acid depletion medium contains 1.7 g/L yeast nitrogen base without ammonium sulfate (#1553 Conda Pronadisa, Spain) and 2% glucose with a mix of 200 mg/L methionine, leucine, histidine, and uracil (Formedium, UK). For microscopy experiments, cells were diluted and grown until mid-logarithmic phase. Cells were transferred to glass-bottom 384-well microscope plates (Matrical Bioscience) coated with concanavalin A (Sigma-Aldrich). After 20 min, wells were washed twice with media to remove non-adherent cells. Re-imaging of contact site screen hits was performed at room temperature (RT) using a VisiScope Confocal Cell Explorer system composed of a Zeiss Yokogawa spinning disk scanning unit (CSU-W1) coupled with an inverted Olympus IX83 microscope (named Micro1 for reference). Single focal plane images were acquired with a 60×oil lens and were captured using a PCO-Edge sCMOS camera, controlled by VisiView software (GFP/Venus at 488 nm, RFP/mCherry at 561 nm, or BFP/CFP at 405 nm). Additional imaging was performed using three additional microscopy systems (Micro 2, 3, and 4; highlighted in figure legends). Micro 2 – an automated inverted fluorescence microscope system (Olympus) harboring a spinning disk high-resolution module (Yokogawa CSU-W1 SoRa confocal scanner with double micro lenses and 50 μm pinholes). Cells were recorded at 30 °C using a 60 X oil lens (NA 1.42) and with a Hamamatsu ORCA-Flash 4.0 camera. Fluorophores were excited by a laser and images were recorded in three channels: GFP (excitation wavelength 488 nm, emission filter 525/50 nm), mCherry (excitation wavelength 561 nm, emission filter 617/73 nm) and DAPI (excitation wavelength 405 nm, emission filter 447/60). Image acquisition was performed using scanR Olympus soft imaging solutions version 3.2. Images were transferred to ImageJ (https://imagej.nih.gov), for slight, linear, adjustments to contrast and brightness. Micro 3 a DMi8 microscope (Leica Microsystems) with a high-contrast Plan Apochromat 63×/1.30 Glyc CORR CS objective (Leica Microsystems), an ORCA-Flash4.0 digital camera (Hamamatsu Photonics) and MetaMorph 7.8 software (MDS Analytical Technologies). Micro 4 – a PerkinElmer Ultraview Vox spinning disk confocal microscope with a 100×CFI Plan Apochromat VC oil-immersion objective lens (1.4 NA), Nikon TiE inverted stand attached to a Yokogawa CSU-X1 spinning disk scan head, a Hamamatsu C9100-13 EMCCD camera, Prior NanoscanZ piezo focus, and a Nikon Perfect Focus System (PFS). All images were collected as square images with 512x512 pixels. The brightness and contrast of images were linearly adjusted and cropped in Photoshop (Adobe) for presentation.

## Organelle labelling and drug treatment

To label LDs, cells were stained with BODIPY 493/503 (Invitrogen) or MDH (Monodansylpentane, also known as AUTODOT). For BODIPY staining, cells were seeded in glass-bottom plates, incubated with 1 µM of BODIPY (dissolved in DMSO) in SD media for 15 min at RT and washed twice before imaging in SD media (control cells were treated with the same concentration of DMSO). For MDH staining, cells were seeded in glass-bottom plates, washed once with PBS and incubated with 100 µM of MDH (dissolved in DMSO) in PBS for 15 min at 30 °C. After incubation, cells were washed twice with PBS and imaged in PBS.

To label the cell periphery with TRITC-ConA (Life Technologies), cells were incubated with the conjugate for 30 min, washed twice with PBS and seeded on a glass-bottom plate prior to imaging.

For drug treatment, cells were grown for 4 hr until mid-log, seeded on glass-bottom plates and imaged once before adding drugs. Amphotericin B (AmB, Invitrogen) was added to SD media at 2 µg/ml and cells were incubated and imaged every 30 min at 30 °C. Fluconazole (Sigma) was added to SD media at 40 µg/ml and cells were incubated and imaged every 30 min at 30 °C.

For nutrient shift experiments, cells were first imaged in minimal synthetic dextrose media, the media aspirated and replaced with PBS with or without glucose supplementation.

## High content contact site protein screens

To identify new potential contact site proteins, query strains containing contact site reporters for six different contact sites were crossed against a subset of the SWAT *TEF2-mCherry* library (1165 strains) (*Weill et al., 2018*; *Yofe et al., 2016*) using the synthetic genetic array method (*Cohen and Schuldiner, 2011*; *Tong and Boone, 2006*). This subset of the SWAT library contains all strains where protein localization is annotated as "punctate", as this is the most common phenotype for contact site proteins, as well as a set of known contact site proteins. Protein punctate localization was based on previous annotations manually performed in our laboratory for three different libraries: *NOP1*-GFP, *TEF2*-mCherry and NATIVEpr-GFP, and the equivalent *TEF2*-mCherry strains annotated as punctate in both NOP1-GFP and NATIVEpr-GFP were also added to the final subset. To perform yeast manipulations, query and library strains were handled in high-density format (384–1536 strains per plate) using a RoToR bench-top colony arraying instrument (Singer Instruments, UK). In short, cells were mated on rich medium plates and diploids were selected in SD$_{MSG}$-His containing Geneticin (200 µg/ml) (Formedium) and Nourseothricin (200 µg/ml) (WERNER BioAgents "ClonNAT"). Sporulation was induced by transferring cells to nitrogen starvation media plates for 8 days. Haploid cells were selected in SD-Leu (for Mat alpha selection) and -Arg-Lys with toxic amino-acid derivatives canavanine and thialysine (Sigma-Aldrich) to select against diploids. Finally, haploid cells containing the combination of manipulations desired were selected using SD$_{MSG}$-His containing Geneticin (200 µg/ml) and Nourseothricin (200 µg/ml), and final libraries containing the genomic traits were created. For each library, a set of strains were verified by microscopy and check PCR.

The new libraries were screened using an automated microscopy setup. Cells were transferred from agar plates into 384-well plates for growth in liquid media using the RoToR arrayer. Liquid cultures were grown in a LiCONiC incubator overnight at 30 °C in SD-His. A JANUS liquid handler (Perkin-Elmer) connected to the incubator was used to dilute the strains to an OD600 of ~0.2 into plates containing SD-His medium, and plates were incubated at 30 °C for 4 hr. Strains were then transferred by the liquid handler into glass-bottom 384-well microscope plates (Matrical Bioscience) coated with Concanavalin A (Sigma-Aldrich). After 20 min, wells were washed twice with SD-Riboflavin media to remove non-adherent cells and to obtain a cell monolayer. The plates were then transferred to an Olympus automated inverted fluorescent microscope system using a robotic swap arm (Hamilton). Cells were imaged in SD-Riboflavin at 18–20 °C using a ×60 air lens (NA 0.9) and with an ORCA-ER charge-coupled device camera (Hamamatsu), using the ScanR software. Images were acquired in two channels: GFP (excitation filter 490/20 nm, emission filter 535/50 nm) and mCherry (excitation filter 572/35 nm, emission filter 632/60 nm). After acquisition, images were manually reviewed using ImageJ.

A strain was considered a 'hit' if the overexpressed protein co-localized with the contact site reporter and/or if it affected the number and/or brightness of the contact site reporter. Co-localization and effect were scored manually. Each strain was given a number based on co-localization: 0, no co-localization; 1, full co-localization – all of the contact site signal co-localizes with the mCherry signal

and vice-versa; 2, partial co-localization green – all the contact site signal (green) co-localizes with the mCherry but not all the mCherry signal co-localizes with the contact site; 3, partial co-localization red – all the mCherry signal (red) co-localizes with the contact site but not all the contact site signal co-localizes with the mCherry; and 4, partial co-localization – some of the mCherry and contact site signals co-localize but some do not co-localize. To assess effect, a threshold for brightness was defined using control strains (only expressing the contact site reporter). Images of the control strain were visually compared against all other strains to select those that showed clear changes in the number of puncta per cell and changes in signal brightness. For comparison, 3 microscope fields of the same strain were used and cell density was taken into consideration by looking at the brightfield channel. Each strain was given an effect letter: L, less contact site puncta; M, more contact site puncta; W, weaker contact site puncta; and B, brighter contact site reporter (0 if no effect). Visual effect assessment was based on overall changes in cells per microscopy field and not that of each individual cells.

Hits form the library were re-picked and re-imaged using a spinning disk confocal microscope (as above, Micro 1) to improve resolution, and only proteins that still fit the co-localization/effect criteria were kept in the final hit list.

The final list of hits for each contact site was manually annotated. As some organelles are smaller than the diffraction limit of confocal microscopy (peroxisomes, lipid droplets and Golgi), we were unable to distinguish proteins localized to the organelle from those localized to the contact sites analyzed. To simplify the hit list, we assembled a list of known proteins for these organelles, and removed them if they were marked as co-localized in the final hit list unless they affected the contact site extent or brightness.

## Protein modelling

Ab initio folding by co-evolution analysis by trRosetta and Alphafold2 was carried out at the Yanglab web server and at the AlphaFold2.ipynb colab respectively (*Mirdita et al., 2021*; *Yang et al., 2020*). Settings were standard, with AlphaFold2 set to return relaxed models (Amber = On) and to ignore templates. For Ypr097w, uninformative regions were omitted in trRosetta 212–244 and 366–465; in AlphaFold2 305–321, 356–468 and 690–780. In the latter case, to model the complete protein, omitted regions were re-added using the new AlphaFold2 structure as the template, by one-to-one threading in Phyre2 (*Kelley et al., 2015*). To model Csf1, since neither it nor any homolog has been published among the proteins modelled by AlphaFold2, 4 overlapping sections (starting with residues 1–1080 and ending with 2070–2959) were predicted by the AlphaFold2.ipynb colab. A single model was created from overlapping portions in Chimera (*Pettersen et al., 2004*). Proteins were visualized in CCP4MG, Chimera (*Pettersen et al., 2004*) and ChimeraX (*Pettersen et al., 2021*) software. Cavity sizes were calculated with MOLE (*Pravda et al., 2018*).

## Phenotype quantification

For quantification of co-localization between the Golgi apparatus and peroxisomes, and to compare co-localization between organelle markers in complete medium or amino acid depletion medium, thousands of cells derived from 3 independent biological repeats were segmented by artificial intelligence algorithms (ScanR Olympus soft imaging solutions, version 3.2). To compare the number of cells with GoPo reporter in complete medium or amino acid depletion medium, 100 cells were manually analyzed from each of 3 biological replicates (total n=3, 300 cells). For quantification of Ypr097w localization, 200 cells (mix of single and budding) were manually analyzed from each of 3 biological replicates (total n=3, 600 cells). Bud size diameter was manually measured in the brightfield channel using ImageJ. For LD quantification and fluconazole effect, 100 cells (mix of single and budding) from 3 independent experiments were manually counted, from a single focal plane (total n=3, 300 cells). For mCherry-D4H localization and distribution, 100 cells (budding) from 3 independent experiments were analyzed (total n=3, 300 cells). To avoid quantification bias, cells were selected in the brightfield channel and then analyzed for GFP or mCherry localization. For the comparison of media effect on GoPo reporter, unpaired t-tests were performed. For multiple comparisons, the ordinary one-way ANOVA test was used with Dunnett's correction for multiple comparisons. ns. not significant, * $p \leq 0.05$, ** $p \leq 0.01$, *** $p \leq 0.001$, **** $p \leq 0.0001$. Graphs and data analysis was performed using GraphPad Prism 9 (GraphPad Software Inc, San Diego, CA).

To calculate the specific PM to cytosol (PM/Cyto) ratio of PS, the relative fluorescence (relative FPM) was quantified as described in *Nishimura et al., 2019*. Briefly, individual cells were chosen from single channel images, lines were drawn across the mother cell and the corresponding fluorescence intensity profiles were plotted. The two highest intensity values, corresponding to signal at the PM, were averaged (FPM). Intensity measurements were also taken from along lines drawn through the cytosol (Fcytosol) and background (Fbackground) and PM relative fluorescence was calculated by using the equation: relative FPM = (FPM-Fbackground)/(Fcytosol-Fbackground). Peaks in intensity profiles were automatically calculated by an Excel VBA macro.

## Serial dilution growth assay

To assess sensitivity to AmB, cells were grown overnight in SD media and back diluted for 4 hr until they reached mid-logarithmic growth. For each strain, cells were diluted to a starting OD600=0.1 and serial 10-fold dilutions were prepared in 96-well plates. 2.5 µl of each dilution were plated in SD agar plates containing 2 µg/ml of AmB (*Figure 5A*) or 100 ng/ml (*Figure 4—figure supplement 2D*) and imaged 3 days after growing at 30 °C. Plates without AmB were prepared in parallel to control for the effect of each genetic manipulation on growth. For each condition, 2 replicate plates were prepared in each of 3 independent experiments.

## Lipid droplet purification

Lipid droplets were purified as previously described (*Athenstaedt, 2010*) with modifications (*Ganesan et al., 2020*). Briefly, yeast cultures were grown for 24 hr in defined medium. Cells were then collected, washed once with water and the wet weight was determined. Cells were shaken at 30 °C for 10 min in 0.1 M Tris/ $H_2SO_4$ buffer (pH 9.4) containing freshly added 10 mM DTT and then washed with 1.2 M sorbitol in 20 mM $KH_2PO_4$ (pH 7.4). Zymolyase 20T (2 mg per gram cell wet weight) was added to obtain spheroplasts (*Daum et al., 1982*). Spheroplasts were washed twice with 1.2 M sorbitol in 20 mM $KH_2PO_4$ (pH 7.4) prior to homogenization. The spheroplasts were then resuspended in buffer A (10 mM MES-Tris (pH 6.9) 12% (w/w) Ficoll 400–0.2 mM EDTA, 1 mM PMSF 1.5 µg/ml pepstatin and Complete EDTA-free protease inhibitor mixture) to a final concentration of 2.5 g per cell wet weight per ml. Spheroplasts were then homogenized using a Dounce homogenizer by employing 30 strokes using a loose-fitting pistil, followed by a centrifugation at 5000 g at 4 °C for 5 min. The resulting supernatant (~5 ml) was transferred into an ultracentrifugation tube, overlaid with an equal volume of buffer A, and centrifuged at 100,000 g at 4 °C for 60 min in a swinging rotor (SW40Ti) (Beckman Coulter OptimaTM L–100 K). Following centrifugation, a white floating layer was collected from the top of the overlay and resuspended gently in buffer A by using a homogenizer with a loose-fitting pistil. The homogenized floating layer was again transferred into an ultracentrifugation tube, overlaid with buffer B (10 mM MES-Tris (pH 6.9) 8% (w/w) Ficoll 400–0.2 mM EDTA) and centrifuged at 100,000 g (4 °C) for 30 min. After the second centrifugation, the top white floating layer containing LDs was removed. A subphase laying beneath the white floating layer was also collected (~0.3 ml). The top white floating layer was then suspended in buffer containing 8% (w/w) Ficoll 400/0.6 M sorbitol and overlaid with buffer containing 0.25 M sorbitol. After a centrifugation at 100,000 g for 30 min, a final white floating layer was collected containing highly purified LDs. The purified LDs were homogenized using a Dounce homogenizer by applying 20 strokes.

## Western blot analysis of lipid droplet fractions

BCA assay (Thermo Scientific) with bovine serum albumin as a standard was used to determine protein concentration. Samples were resuspended in gel loading buffer (0.2 M Tris HCl (pH 6.8), 8% SDS, 0.4% bromophenol blue, and 40% glycerol) and boiled for 1 min. SDS-PAGE was carried out by the Laemmli method (*Laemmli, 1970*). Proteins were separated by 8% resolving gel containing trichloroethanol (TCE, Sigma) to visualize proteins (*Ladner et al., 2004*). Proteins were transferred to a polyvinylidene fluoride (PVDF) membrane (Millipore) using a Bio-Rad transfer system at 90 V for 80 min, and then stained with Red Ponceau (Sigma) to monitor the transfer. For western blot analysis, monoclonal anti-GFP antibody (Roche) and polyclonal antibody raised against Erg6 (kind gift from G. Daum, Universität Graz) were used and subsequently with horseradish peroxidase-conjugated secondary antibodies (Invitrogen, Thermo Fisher). Enhanced chemiluminescence (Amersham, GE Healthcare) was used for detection followed by imaging (Amersham Imager 600).

### Western blot analysis of Ypr097w-GFP mutants

Yeast cells were grown to log phase in SD media at 30 °C and 10 OD600/ml equivalents of cells were harvested and stored at –80 °C. Cells were thawed and lysed by vortexing in 100 µl of Thorner buffer (8 M Urea, 5% SDS, 40 mM Tris-Cl (pH 6.4), 1% beta-mercaptoethanol and 0.4 mg/ml bromophenol blue) with ~100 µl of acid-washed glass beads/sample at 70 °C for 5 min. Lysates were centrifuged at 14,000 RPM for 30 s and separated on 8% SDS-PAGE gels followed by western blotting with monoclonal mouse anti-GFP antibodies (11–814–460-001; Roche) or monoclonal mouse anti-PGK1 antibodies (AB_2532235, 22C5D8, Invitrogen), and secondary polyclonal goat anti-mouse antibodies conjugated to horseradish peroxidase (115–035-146; Jackson ImmunoResearch Laboratories). Blots were developed with Amersham ECL (RPN2232, Cytiva) chemiluminescent western blot detection reagent and exposed using Amersham Hyperfilm ECL (GE Healthcare).

### Lipid extraction and thin layer chromatography

The indicated yeast cultures were grown for 24 hr in synthetic defined medium. Cultures were then diluted and allowed to grow into mid-log phase (5 hr), early stationary phase for an additional 24 hr or additional 48 hr for late stationary phase. For growth resumption samples, cells were pelleted after growing for 24 hr, resuspended in fresh medium and cultured for an additional 45 min. For each of the growth phases, 20 OD600 of cells were collected. Lipid extractions were performed as previously described (*Zaremberg and McMaster, 2002*). Briefly, cell pellets were resuspended in 1 ml of CHCl$_3$/CH$_3$OH (1/1), and disrupted with 0.5 mm acid-washed glass beads for 1 min at 4 °C using a BioSpec Multi-Bead Beater. The supernatant was transferred into a glass tube and the beads were washed with 1.5 ml of CHCl$_3$/CH$_3$OH (2/1, v/v) and transferred into the same glass tube. A total of 1.5 ml of water and 0.5 ml of CHCl$_3$ was added to the glass tube to facilitate phase separation. The organic phase containing the lipids was collected after a 10 min centrifugation at 2500 g. A total of 2.5 ml of artificial aqueous layer, CHCl$_3$/CH$_3$OH/H$_2$0 (3/48/47, v/v/v) was then added to the organic phase and subjected to another 10 min centrifugation at 2500 g. The final organic phase was collected and dried using nitrogen gas. Neutral lipids were analyzed by thin layer chromatography using Silica Gel on TLC aluminum foil plates (Sigma) and separated using solvent system I: petroleum ether/diethyl ether/acetic acid (80/20/1, v/v/v). Lipids were resuspended in CHCl$_3$ and equal amount of lipids were spotted on the plates. Lipids were visualized using iodine vapor staining or by dipping the plates in a solution containing 3.2% H$_2$SO$_4$, 0.5% MnCl$_2$, 50% EtOH and charring the plates at 120 °C for 30 min. Plates were imaged using an Amersham Imager 600. Neutral lipid quantification was done using the linear range of lipid standards and ImageJ. GraphPad Prism 5 software was used for statistical analysis of data and preparation of figures.

Lipid standards used were purchased from Sigma: cholesteryl-ester (700222), 1–2-dioleoyl-sn-glycerol (800811), ergosterol (E6510), oleic acid (O1008), squalene (S3626), and triolein (870110).

### Cell culture and harvest protocol for lipidomics

A single colony from an agar plate with synthetic complete dextrose (SCD) medium was used to inoculate a pre-culture of 3 ml in liquid SCD medium. After cultivation under constant agitation at 30 °C for 19 h, stationary cells were used to inoculate a main culture to an OD600 of 0.1. Logarithmically growing cells were harvested at an OD600 of 1.0, while stationary cells were harvested 48 h after inoculation of the main culture. 20 OD units of cells were harvested by centrifugation (3500 g, 5 min, 4 °C), followed by three washing steps in ice-cold buffer containing 155 mM ammonium bicarbonate and 10 mM sodium azide using rapid centrifugation (10,000 g, 20 s, 4 °C). The final cell pellet was snap-frozen with liquid nitrogen and stored at –80 °C until cell lysis. To this end, the cell pellets were thawed on ice and then resuspended in 1 ml of ice-cold 155 mM ammonium bicarbonate. The suspension was transferred into a fresh 1,5 ml reaction tube containing 200 µl of zirconia beads (0,5 mm bead size). Cell disruption was induced by vigorous shaking using a DisruptorGenie for 10 min at 4 °C. 500 µl of the resulting lysate was snap-frozen and sent to Lipotype GmbH for lipid extraction and further analysis.

### Lipid extraction for mass spectrometry (lipidomics)

Mass spectrometry-based lipid analysis was performed by Lipotype GmbH (Dresden, Germany) as described (*Ejsing et al., 2009*; *Klose et al., 2012*). Lipids were extracted using a two-step chloroform/

methanol procedure (*Ejsing et al., 2009*). Samples were spiked with internal lipid standard mixture containing: CDP-DAG 17:0/18:1, cardiolipin 14:0/14:0/14:0/14:0 (CL), ceramide 18:1;2/17:0 (Cer), diacylglycerol 17:0/17:0 (DAG), lyso-phosphatidate 17:0 (LPA), lyso-phosphatidyl- choline 12:0 (LPC), lyso-phosphatidylethanolamine 17:1 (LPE), lyso- phosphatidylinositol 17:1 (LPI), lyso-phosphatidylserine 17:1 (LPS), phosphatidate 17:0/14:1 (PA), phosphatidylcholine 17:0/14:1 (PC), phosphatidyletha-nolamine 17:0/14:1 (PE), phosphatidylglycerol 17:0/14:1 (PG), phosphatidylinositol 17:0/14:1 (PI), phosphatidylserine 17:0/14:1 (PS), ergosterol ester 13:0 (EE), triacylglycerol 17:0/17:0/17:0 (TAG), stigmastatrienol, inositolphosphorylceramide 44:0;2 (IPC), mannosyl-inositolphosphorylceramide 44:0;2 (MIPC) and mannosyl-di- (inositolphosphoryl)ceramide 44:0;2 (M(IP)2 C). After extraction, the organic phase was transferred to an infusion plate and dried in a speed vacuum concentrator. 1st step dry extract was re-suspended in 7.5 mM ammonium acetate in chloroform/methanol/propanol (1:2:4, V:V:V) and 2nd step dry extract in 33% ethanol solution of methylamine in chloroform/methanol (0.003:5:1; V:V:V). All liquid handling steps were performed using Hamilton Robotics STARlet robotic platform with the Anti Droplet Control feature for organic solvents pipetting.

## Mass spectrometry data acquisition (lipidomics)

Samples were analyzed by direct infusion on a QExactive mass spectrometer (Thermo Scientific) equipped with a TriVersa NanoMate ion source (Advion Biosciences). Samples were analyzed in both positive and negative ion modes with a resolution of $Rm/z=200 = 280,000$ for MS and $Rm/z=200 = 17,500$ for MSMS experiments, in a single acquisition. MSMS was triggered by an inclusion list encompassing corresponding MS mass ranges scanned in 1 Da increments (*Surma et al., 2015*). Both MS and MSMS data were combined to monitor EE, DAG and TAG ions as ammonium adducts; PC as an acetate adduct; and CL, PA, PE, PG, PI, and PS as deprotonated anions. MS only was used to monitor LPA, LPE, LPI, LPS, IPC, MIPC, M(IP)2 C as deprotonated anions; Cer and LPC as acetate adducts and ergosterol as protonated ion of an acetylated derivative (*Liebisch et al., 2006*).

## Data analysis and post-processing (lipidomics)

Data were analyzed with in-house developed lipid identification software based on LipidXplorer (*Herzog et al., 2012*; *Herzog et al., 2011*). Data post-processing and normalization were performed using an in-house developed data management system. Only lipid identifications with a signal-to-noise ratio >5, and a signal intensity 5-fold higher than in corresponding blank samples were considered for further data analysis.

For statistical analysis of lipidomics data, the mol % of each lipid class was calculated against the total sum of lipids per sample. For each metabolic state (mid-logarithmic growth and stationary), 3 biological replicates of WT vs *THD3pr-YPR097W* strains were compared using multiple unpaired t-tests, assuming Gaussian distribution. Data was corrected for False Discovery Rate using the two-stage step-up method of Benjamini, Krieger and Yekutieli (Q=5%). Data analysis was performed using GraphPad Prism 9 (GraphPad Software Inc, San Diego, CA).

## DHFR protein fragment complementation assay and ontology analysis

Endogenous YPR097W was C-terminally fused to one half of a methotrexate-resistant variant of the essential DHFR enzyme in a MATa strain, which was crossed into a library of MATα strains (n = ~4300) expressing proteins fused to the complementary half of the DHFR enzyme (*Tarassov et al., 2008*). Diploids were subjected to two rounds of double mutant selection followed by two rounds of selection on media containing 200 µg/µl methotrexate (MTX). All yeast manipulations were carried out using the BM3-BC microbial pinning robot (S&P Robotics Inc, Canada) with a 384-pin tool. Colonies were maintained in either 768- or 1536-density and technical duplicates were pinned for MTX selection steps. Colony area was analyzed using CellProfiler (*Lamprecht et al., 2007*), after 7 days at 30 °C. Interactions with abundant, cytosolic proteins were filtered out by comparing colony area of Ypr097w DHFR diploids to DHFR diploids containing an overproduced DHFR bait fragment fused to a cytosolic reporter. Z-scores were generated using median colony area from two technical replicate for each Ypr097w-prey combination. Functional analysis of Ypr097w DHFR interactors (Z>2.5) was performed using the Gene Ontology (*Ashburner et al., 2000*; *The Gene Ontology Consortium, 2019*) GO Enrichment Analysis tool (*Mi et al., 2019*).

## Acknowledgements

We wish to thank Einat Zalckvar and Maria Bohnert for enriching scientific discussions, many great ideas and critical reading of this manuscript. We thank Yoav Peleg for cloning of the phosphomutant plasmids. We thank Ron Rotkopf for statistical analysis support. We thank Christian Landry for providing the C-terminally tagged DHFR prey library and plasmid for generation of a bait Ypr097w construct. We thank Joel Goodman, Sophie G Martin, Won-Ki Huh, David Breslow, Naama Barkai and Vlad Denic for plasmids.

RE is funded by the European Research Council under the European Union's Horizon 2020 research and innovation program (grant agreement no. 866011). Collaborative work between the Ernst and Schuldiner labs is supported by a Volkswagen Foundation "Life" grant (93089 and 93092). Work in the Schuldiner lab is also supported by a Deutsche Forschungsgemeinschaft (DFG) SFB1190 grant. The robotic system of the Schuldiner lab was purchased through the kind support of the Blythe Brenden-Mann Foundation. MS is an incumbent of the Dr. Gilbert Omenn and Martha Darling Professorial Chair in Molecular Genetics. IGC is a recipient of an EMBO Long-term Fellowship (ALTF-580–2017). CJS acknowledges support by MRC funding to the MRC LMCB University Unit at UCL, award code MC_UU_00012/6. TL was funded by the BBSRC UK (Grant BB/M011801/1). The Conibear lab acknowledges support by the Canada Foundation for Innovation (Leading Edge Fund 30636); Canadian Institutes of Health Research (grant 148756 to EC, CGS-M Frederick Banting and Charles Best Canada Graduate Scholarship to SKD and SPS, CGS-D Frederick Banting and Charles Best Canada Graduate Scholarship to SKD); Natural Sciences and Engineering Research Council of Canada (PGS-D to SPS); BC Children's Hospital Research Institute (Sue Carruthers Graduate Studentship to SKD and Jan M Friedman Graduate Studentship to SPS) and University of British Columbia 4 Year Doctoral Fellowship to SKD and SPS. VZ is supported by the Natural Sciences and Engineering Research Council of Canada (NSERC).

## Additional information

### Competing interests

Maya Schuldiner: Reviewing editor, eLife. The other authors declare that no competing interests exist.

### Funding

| Funder | Grant reference number | Author |
|---|---|---|
| European Molecular Biology Organization | ALTF-580-2017 | Inês Gomes Castro |
| Volkswagen Foundation | 93092 | Robert Ernst<br>Maya Schuldiner |
| Deutsche Forschungsgemeinschaft | SFB1190 | Maya Schuldiner |
| Medical Research Council | MC_UU_00012/6 | Christopher Stefan |
| Biotechnology and Biological Sciences Research Council | Grant BB/M011801/1 | Tim P Levine |
| Canada Foundation for Innovation | 30636 | Elizabeth Conibear |
| Canadian Institutes of Health Research | 148756 | Elizabeth Conibear |
| Canadian Institutes of Health Research | CGS-M Frederick Banting and Charles Best Canada Graduate Scholarship | Samantha Katarzyna Dziurdzik<br>Shawn P Shortill |
| Canadian Institutes of Health Research | CGS-D Frederick Banting and Charles Best Canada Graduate Scholarship | Samantha Katarzyna Dziurdzik |

| Funder | Grant reference number | Author |
| --- | --- | --- |
| Natural Sciences and Engineering Research Council of Canada | PGS-D | Shawn P Shortill |
| BC Children's Hospital | Sue Carruthers Graduate Studentship | Samantha Katarzyna Dziurdzik |
| BC Children's Hospital | Jan M. Friedman Graduate Studentship | Shawn P Shortill |
| University of British Columbia | Doctoral Fellowship | Shawn P Shortill Samantha Katarzyna Dziurdzik |
| Natural Sciences and Engineering Research Council of Canada | PIN303585-2016 | Vanina Zaremberg |
| European Commission | ERC-CoG 866011 | Robert Ernst |

The funders had no role in study design, data collection and interpretation, or the decision to submit the work for publication.

## Author contributions

Inês Gomes Castro, Conceptualization, Data curation, Formal analysis, Supervision, Validation, Investigation, Visualization, Methodology, Writing – original draft, Writing – review and editing; Shawn P Shortill, Conceptualization, Data curation, Formal analysis, Validation, Investigation, Visualization, Methodology, Writing – review and editing; Samantha Katarzyna Dziurdzik, Conceptualization, Formal analysis, Validation, Investigation, Visualization, Methodology, Writing – review and editing; Angela Cadou, Conceptualization, Formal analysis, Validation, Investigation, Visualization, Writing – review and editing; Suriakarthiga Ganesan, Data curation, Formal analysis, Validation, Investigation, Methodology, Writing – review and editing; Rosario Valenti, Formal analysis, Visualization; Yotam David, Michael Davey, Investigation; Carsten Mattes, Formal analysis, Validation, Investigation, Writing – review and editing; Ffion B Thomas, Formal analysis, Investigation; Reut Ester Avraham, Hadar Meyer, Validation, Investigation; Amir Fadel, Software, Formal analysis; Emma J Fenech, Investigation, Writing – review and editing; Robert Ernst, Data curation, Supervision, Funding acquisition, Writing – review and editing; Vanina Zaremberg, Conceptualization, Resources, Supervision, Funding acquisition, Investigation, Visualization, Project administration, Writing – review and editing; Tim P Levine, Conceptualization, Funding acquisition, Validation, Investigation, Visualization, Methodology, Writing – original draft, Writing – review and editing; Christopher Stefan, Conceptualization, Supervision, Funding acquisition, Project administration, Writing – review and editing; Elizabeth Conibear, Conceptualization, Supervision, Funding acquisition, Methodology, Project administration, Writing – review and editing; Maya Schuldiner, Conceptualization, Supervision, Funding acquisition, Writing – original draft, Project administration, Writing – review and editing

## Author ORCIDs

Inês Gomes Castro http://orcid.org/0000-0003-4669-985X
Shawn P Shortill http://orcid.org/0000-0001-8742-7442
Samantha Katarzyna Dziurdzik http://orcid.org/0000-0002-3553-1595
Suriakarthiga Ganesan http://orcid.org/0000-0001-7430-7740
Rosario Valenti http://orcid.org/0000-0002-6093-1873
Yotam David http://orcid.org/0000-0003-4992-7045
Michael Davey http://orcid.org/0000-0002-1172-5934
Ffion B Thomas http://orcid.org/0000-0002-1899-1397
Hadar Meyer http://orcid.org/0000-0001-5725-0447
Emma J Fenech http://orcid.org/0000-0003-4414-3233
Robert Ernst http://orcid.org/0000-0002-0283-3490
Vanina Zaremberg http://orcid.org/0000-0001-7009-2172
Tim P Levine http://orcid.org/0000-0002-7231-0775
Christopher Stefan http://orcid.org/0000-0002-4118-5721
Elizabeth Conibear http://orcid.org/0000-0001-5129-0499
Maya Schuldiner http://orcid.org/0000-0001-9947-115X

Decision letter and Author response
Decision letter https://doi.org/10.7554/eLife.74602.sa1
Author response https://doi.org/10.7554/eLife.74602.sa2

## Additional files

### Supplementary files

• Supplementary file 1. List of strains in the "Puncta Library" collection. List of the subset of strains from the SWAT *TEF2pr*-mCherry library selected for this work based on protein localization in cells.

• Supplementary file 2. Newly suggested contact site residents and regulators. List of hits for each of the 6 screens performed and their phenotype based on co-localization with contact site reporter and/ effect over the contact site reporter.

• Supplementary file 3. Ypr097w DHFR Proximity Interactome. List of noise-filtered Ypr097w DHFR interactors (Z-Score >2.5). Hits with reported lipid droplet localization (data from *Huh et al., 2003*; *Weill et al., 2018*) or membership in a validated LD proteome (data from *Currie et al., 2014*) are colored pink, while hits with reported ER localization that are not present in the validated LD proteome are colored blue.

• Supplementary file 4. Lipidomic analysis of WT and *TDH3pr-YPR097W* strains.

• Supplementary file 5. *S. cerevisiae* strains used in this study.

• Supplementary file 6. Plasmids used in this study.

• Supplementary file 7. Primers used in this study.

• Transparent reporting form

### Data availability

Raw numerical data and statistical results used to generate Figures 1B, 1S1B, 1S1D, 3C, 3D, 3S1C, 5D, 5S1A and 5S2B can be found as Source Data excel files with the name of the respective figure. DHFR raw data and GO term enrichment data can be found in "Supplementary File 3_Lec1 DHFR Hits". Lipidomics raw data and statistical results can be found in "Supplementary File 4_Lipidomics".

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
