## [Editor Report]

This manuscript describes an extensive and systematic analysis of membrane contact sites in budding yeast to uncover novel proteins required for tethering organelles and modulation of membrane contacts. The authors identify over 100 new potential contact site proteins and effectors including proteins associated with the recently discovered plasma membrane-LD (pClip) and Golgi-peroxisome (GoPo) contact sites. Further, the authors identify and characterize novel lipid transport proteins associated with the pClip as well as Lec1, an ER-Lipid droplet contact site associated protein that contains a novel putative lipid-binding domain and may facilitate ergosterol transport between the plasma membrane and lipid droplets.

---

## [Decision Letter]

**Decision letter after peer review:**

Thank you for submitting your article "Systematic analysis of membrane contact sites in *Saccharomyces cerevisiae* uncovers modulators of cellular lipid distribution" for consideration by *eLife*. Your article has been reviewed by 3 peer reviewers, one of whom is a member of our Board of Reviewing Editiors, and the evaluation has been overseen by a Reviewing Editor and Vivek Malhotra as the Senior Editor. The reviewers have opted to remain anonymous.

Essential revisions:

1) Better characterisation of the pCLIP and GoPo reporters and the respective contact sites.

2) The identification of Ypr097W/Lec1 as a putative lipid binding protein and its effects in ergosterol distribution are very interesting. However, the claim that Lec1 is part of a new family of lipid binding proteins requires further support, such as demonstration of its lipid binding activity and/or use of mutagenesis to show a role of the hydrophobic pocket.

3) The localization of Hob1, Hob2 and Csf1 to ER-PM MCS should be further characterized. In particular, it should include the analysis of C-terminally tagged endogenous proteins together with organelle markers.

*Reviewer #1 (Recommendations for the authors):*

1) Validation of the MCS reporters is not shown. This is particularly important for pCLIP and GoPo, which have not been reported before. Fluorescence complementation between two proteins that normally localize to different organelles is far from demonstrating the existence of a MCS between those organelles. It would be important to demonstrate using marker proteins and ideally electron microscopy/CLEM the existence of the mentioned MCS and the suitability of the fluorescent reporter.

2) As pointed out above, the identification of a phenotype in ergosterol distribution for Ypr097W/Lec1 is very interesting. However, it is unclear how this observation relates with the localization of Lec1 to LDs, which is observed only upon over-expression. Instead, further characterization of Ypr097w phenotype (via mutagenesis, modulation of ergosterol biosynthetic pathway, test ability to bind ergosterol, etc) in ergosterol distribution would be a plus.

3) The description of phenotypes is mostly qualitative. For example, when describing effectors, it is unclear how: 1) full vs partial localization was scored; 2) the brightness of the punctae. Was is there a threshold for these effectors? all punctae became brighter, how much brighter? or only a fraction? some details on the criteria would be useful. Also, Ypr097w was described to localize "primarily at buds and necks….". Simple analysis of the images appears to suggest that the localization to buds and bud necks depend on the bud size. Along the same line is stated that "some increase in the number of internal puncta". Again, it would be good to provide a quantitative description of the phenotypes described throughout the manuscript.

4) In some MCS there is the enrichment of a protein from a third organelle (for example Nvj1 is enriched at Po-Mito or Vac8 at LD-PM). Is this because a third organelle is part of the MCS or is it a false positive? Please comment. In addition, several proteins appear to be present in most MCS. This behaviour has been described for soluble MCS proteins such as Vps13. However, is this also expected for membrane proteins like Mdm1 and Tul1?

5) Both pCLIP and Lider involve LDs. Since the imaging of these organelles is diffraction limited, is it expected that overexpression of any protein that impacts in LD size would, in principle, result in changes in the size/brightness of the Lider and pCLIP reporters?

*Reviewer #2 (Recommendations for the authors):*

1. The identification of Ypr097w/Lec1 is extremely interesting. However, more can be done on this protein. The large cavity of Lec1 is unique, and is suggested to host tiny LDs/TAG and PC mixtures. It would be important to test this. For instance, does Lec1 bind TAG or sterol esters or PC? See PMID 31708432 for relevant assays. If so, some sort of lipid transfer assay in vitro would significantly strengthen this work.

2. The D4H phenotype does not necessarily indicate the Lec1's primary function is in sterol trafficking/metabolism. Latest work has shown that phosphatidylserine can greatly impact D4H/sterol distribution. See PMID 33929485 and 32690708. It would be good to also examine the cellular distribution of PS.

3. Figure 3C is confusing. Can the authors do similar experiment as 3F but with endogenous promoter and under the same condition as 3C?

4. Not clear to this reviewer how specific Osw5/Ldo16 is to the LDs. Few proteins are exclusively localized to LDs. Not sure if there is such a protein at all in yeast. Perhaps some past validation work can be made clearer.

*Reviewer #3 (Recommendations for the authors):*

1. In Figure 1 the authors use a set of split venus probes to detect specific MCSs primarily utilizing constructs characterized in their previous publication (Shai et. al, Nature Communications, 2018). They also introduce a new probe for detection of Golgi-Peroxisome membrane contact sites (GoPo). Proper localization of the GoPo probe needs to be demonstrated to validate this data.

2. In Figure 2 the authors identify 3 proteins (Fmp27, Ypr117w, and Csf1) that colocalize with the pCLIP. The authors note that these proteins have previously been shown to localize to ER-PM sites when tagged at their C-terminus. They propose that by tagging these proteins at the N-terminus they have disrupted their endogenous localization to the ER membrane (all 3 have shown or predicted N-terminal ER transmembrane anchors) which has in part led to pCLIP colocalization. The disruption of proper ER localization by these proteins weakens their conclusions that these proteins function at the pCLIP.

For example, an alternative explanation is that N-terminal tagging leads to misfolding of the protein and therefore localization to the LD, as the LD has recently been shown to store misfolded proteins. These proteins may maintain their endogenous affinity for the plasma membrane and thereby localize to the pCLIP.

The authors note that C-terminal tagging of these proteins maintains targeting to the ER and PM as they are found predominantly at ER-PM contact sites. If these proteins truly function at the pCLIP, one would expect that overexpression of C-terminally tagged constructs would show pCLIP colocalization in addition to ER-PM colocalization. The authors should test the localization of at least one C-terminally tagged pCLIP protein and quantify localization to pCLIP vs. ER-PM contacts. Without this data, the conclusion than Fmp27, Ypr117w, and Scf1 are pCLIP proteins is not well supported.

3. Through in silico approaches the authors demonstrate that Lec1 appears to form a large hydrophobic cavity structure similar to predictions of Mdm1 and Nvj3. They propose that these proteins are a new family of lipid binding proteins. However, as the authors did not directly assess the ability of Lec1 to bind lipids they have only shown this is a putative lipid binding protein and the tittle of this section needs to be adjusted accordingly.

Suggestion for future studies:

Due to the small size of organelles involved in the assessed MCSs (lipid droplets, Golgi, and peroxisomes) the authors exclude hits that only demonstrated colocalization and are known resident proteins of these small organelles. While this greatly reduces the potential for false positive hits, it also likely leads to false negatives. A secondary screen focused on this subset of proteins to identify those which may serve as tethers would provide valuable insight in a future study.

---

## [Author Response]

Essential revisions:1) Better characterisation of the pCLIP and GoPo reporters and the respective contact sites.

We have added the characterization of both contact sites as requested.

For the pCLIP – Since this reporter has previously been reported (Shai et al., 2018 Nat Commun 9, 1761. doi:10.1038/s41467-018-03957-8) we have only added one new figure (Figure 2 S1A) which shows the co-localization of the contact site reporter with a lipid droplet marker (MDH) and a cell periphery marker (TRITC-ConA).

For the GoPO – Since this reporter is presented for the first time, we have rigorously characterized the GoPo reporter by looking at the frequency of co-localization between peroxisomes and the Golgi in cells (Figure 1 S1B), the co-localization of the contact site reporter with a peroxisome marker (CFP-SKL) and a Golgi marker (Sec7-mCherry) (Figure 1 S1C), and by identifying a condition where this contact site is increased (Figure 1 S1D).

2) The identification of Ypr097W/Lec1 as a putative lipid binding protein and its effects in ergosterol distribution are very interesting. However, the claim that Lec1 is part of a new family of lipid binding proteins requires further support, such as demonstration of its lipid binding activity and/or use of mutagenesis to show a role of the hydrophobic pocket.

We have attempted to address this issue extensively by creating many strains with multiple mutations at the proposed lipid binding cavity surface of Lec1, and subsequently analyzing protein activity (detailed experimental design, results and conclusions in point-by-point response to reviewers below). Despite the enormous time and resource investment in creation of a huge array of mutants and careful investigation of phenotypes, we were unable to determine unequivocally a clear role for the hydrophobic pocket since all mutations also affected protein stability and/or localization.

However, we would like to point out that a recent publication by Paul and colleagues (Paul et al., 2022 Front. Cell Dev. Biol. 116. doi:10.3389/fcell.2022.826688) analyzed the structure of Lec1 and additional PX-domain containing proteins using a similar approach to ours. Taking into consideration the known functions of the PX-domain containing proteins and the large hydrophobic cavity identified in these proteins, the authors reached the same conclusion as us, suggesting that these represent a new family of lipid binding/transporting proteins. In addition, although the Paul et al., study includes crystallography data of other domains that do not participate in the proposed cavity, the authors were unable to purify full SNX proteins to be able to perform biochemical assays, and we expect similar difficulties would be the case for Lec1. For this reason, we have toned down our wording to state very carefully that this is only a hypothesis and that clearly future biochemical proof will be required.

3) The localization of Hob1, Hob2 and Csf1 to ER-PM MCS should be further characterized. In particular, it should include the analysis of C-terminally tagged endogenous proteins together with organelle markers.

During the lengthy revision process of our manuscript, C-terminally tagged Hob1, Hob2 and Csf1 have already been shown to localize to ER-Plasma membrane and ER-Mitochondria contact sites (Neuman et al., 2022 J. Cell Sci. 135(5):jcs259086. doi:10.1242/jcs.259086; Toulmay et al., 2022 J. Cell Biol. 221 (3): e202111095. doi:10.1083/jcb.202111095). However, it should be noted that both Hob2 and Csf1 are expressed at very low levels endogenously and therefore had to be tagged with multiple GFP molecules for detection at these contact sites (Toulmay et al., 2022 J. Cell Biol. 221 (3): e202111095. doi:10.1083/jcb.202111095). We now discuss these manuscripts in ours and did not feel we should redo these published experiments

Reviewer #1 (Recommendations for the authors):1) Validation of the MCS reporters is not shown. This is particularly important for pCLIP and GoPo, which have not been reported before. Fluorescence complementation between two proteins that normally localize to different organelles is far from demonstrating the existence of a MCS between those organelles. It would be important to demonstrate using marker proteins and ideally electron microscopy/CLEM the existence of the mentioned MCS and the suitability of the fluorescent reporter.

We thank the reviewer for pointing this out and have now added supplementary characterization of the pCLIP and GoPo contact sites. The pCLIP has been previously described by us (Shai et al. 2018 Nat Commun 9, 1761. doi:10.1038/s41467-018-03957-8) and so we have only added one new figure (Figure 2 S1A) which shows the co-localization of the contact site reporter with a LD marker (MDH) and a cell periphery marker (TRITC-ConA). For the GoPo, since this is the first demonstration of a reporter for this contact site, we have rigorously characterized it by looking at the frequency of co-localization between peroxisomes and the Golgi in the absence of the reporter (Figure 1 S1B), the co-localization of the contact site reporter with a peroxisome marker (CFP-SKL) and a Golgi marker (Sec7-mCherry) (Figure 1 S1C), and by identifying a condition where this contact site is increased (Figure 1 S1D).

Since all supported their function as bone-fide reporters and since performing electron microscopy experiments on these reporters was not possible for us at this time and has not been the standard in the field for other reporters, we hope that this is satisfactory.

2) As pointed out above, the identification of a phenotype in ergosterol distribution for Ypr097W/Lec1 is very interesting. However, it is unclear how this observation relates with the localization of Lec1 to LDs, which is observed only upon over-expression.

We would like to clarify that at endogenous levels Lec1 also localizes to LDs. However, this localization is less pronounced. To clarify this in the text and show this experimentally we have now added an example of the endogenous GFP-tagged protein with the LD marker Faa4-mCherry (Figure 3 S1B), and added a section in the text.

Instead, further characterization of Ypr097w phenotype (via mutagenesis, modulation of ergosterol biosynthetic pathway, test ability to bind ergosterol, etc) in ergosterol distribution would be a plus.

To further characterize the Lec1 phenotype, we looked at changes in ADHpr-GFP-Lec1 localization in cells treated with 40 µg/ml of fluconazole for 3h (Figure 5 S2B-C). Fluconazole is a known inhibitor of Erg11 and treatment with this drug strongly reduces the overall levels of cellular ergosterol, which can be clearly observed by the loss of binding of mCherry-D4H to the plasma membrane (cytosolic signal) (Figure 5 S2B lower right panels). After 3h of treatment with fluconazole, there is a small increase in the number of cells with bud/ bud neck localization for GFP-Lec1. The GFP-Lec1 signal in these cells generally appears brighter than in untreated cells, suggesting that loss of ergosterol potentiates Lec1 accumulation at the bud / bud neck. This result suggests that Lec1 cellular localization is affected by the levels of ergosterol. However, since treatment with high concentration of fluconazole leads to growth arrest (Zhang et al. 2010. PLOS Pathogens 6:e1000939. doi:10.1371/journal.ppat.1000939), it is also possible that this signal increase is the result of Lec1 accumulation at the bud due to a stalling in budding. We now discuss this in the text.

We have also extensively mutagenized Lec1 as requested in an attempt to find a mutant that is still localized to LDs and stable yet not causing sterol redistribution. However, despite great efforts this has proven to be challenging (See below in detailed response to this request from reviewer #2).

3) The description of phenotypes is mostly qualitative. For example, when describing effectors, it is unclear how: 1) full vs partial localization was scored; 2) the brightness of the punctae. Was is there a threshold for these effectors? all punctae became brighter, how much brighter? or only a fraction? some details on the criteria would be useful.

We agree with the reviewer and added information regarding these criteria to the Materials and methods section, subsection “High content contact site protein screens”.

Also, Ypr097w was described to localize "primarily at buds and necks….". Simple analysis of the images appears to suggest that the localization to buds and bud necks depend on the bud size. Along the same line is stated that "some increase in the number of internal puncta". Again, it would be good to provide a quantitative description of the phenotypes described throughout the manuscript.

We agree with the reviewer and have added detailed localization data taking into consideration the bud size (Figure 3 S1C).

4) In some MCS there is the enrichment of a protein from a third organelle (for example Nvj1 is enriched at Po-Mito or Vac8 at LD-PM). Is this because a third organelle is part of the MCS or is it a false positive? Please comment.

This is a pertinent question and indeed, we believe that there is the possibility of 3-way contact sites, and these could potentially be detected when observing the contact between 2 of the 3 organelles due to the diffraction limit of optical microscopy.

To further explore the possibility of 3-way contacts, we first looked at reporters where some hits from an unexpected organelle showed up. To this end, we examined strains expressing either the LiDER, the PerVale or the GoPo, in combination with a vacuole marker (FM4-64) or a mitochondrial marker (MitoTracker). Indeed, we found that the LiDER reporter can localize in the proximity of both the vacuole and mitochondria but only in rare cases. Similarly, the other contacts also show very low to no co-localization with the markers, suggesting that our approach is likely best suited to detect 2-way contacts.

We also looked at the LD-PM contact site reporter with a vacuole marker (FM4-64) as the reviewer requested although Vac8, in our screen, does not co-localize with the contact site reporter, but instead increases the brightness and number of contacts detected in these cells (which can be indirect and does not necessarily suggest a three-way contact). Likewise, in our screen Nvj1 is not enriched at the PO-MITO contact site, but instead decreases the number and brightness of detected contacts in these cells. Indeed, as expected in this strain, the vacuole was not found in proximity to the LD-PM contact site.

At this stage, we have not added these experiments, which only have negative data, to the manuscript and would like to leave it at the discretion of the editor whether to do so.

In addition, several proteins appear to be present in most MCS. This behaviour has been described for soluble MCS proteins such as Vps13. However, is this also expected for membrane proteins like Mdm1 and Tul1?

Some membrane proteins such as the LAM/LTC family of proteins have been shown to be able to localize to multiple contact sites despite having a transmembrane domain hence we think that this is indeed possible. In our data, Mdm1 has been previously shown to localize to LiDER and NVJ contact sites; additionally we shown that it co-localizes with the pCLIP contact site reporter, which is in agreement with its *Drosophila* counterpart (Snazarus) which localizes at ER-LD-PM contacts; additionally, it affects the PerMit reporter but is not localized to it. Tul1 affects/co-localizes with most contacts which may be interesting to follow up on. However, due to the high throughput nature of these experiments, we are unable to follow up on every single protein. We have gone in depth into the Vps13 family of proteins which also localize to multiple contact sites.

5) Both pCLIP and Lider involve LDs. Since the imaging of these organelles is diffraction limited, is it expected that overexpression of any protein that impacts in LD size would, in principle, result in changes in the size/brightness of the Lider and pCLIP reporters?

We agree with the reviewer about this important technical point and its potential impact on our dataset. Hence, we have re-analyzed a subset of the hits for both the pCLIP and LiDER (Author response table 1). We found that in some cases there was as increase in the number/size of LDs as a result of protein overexpression/tagging, but that this was not always the case (approximately true for only 50% of hits analyzed). Protein overexpression and tagging is inherently linked to changes in cellular dynamics and this is something we keep in mind when working with high throughput datasets. However, we are unable to account for all variables when dealing with large datasets, and have therefore not analyzed organelle number and size when determining if a protein constituted a hit in each screen. We have now clearly stated this in the text so that people are aware of this caveat.

**Author response table 1. sa2table1:** 

Contact site	ORF	Gene	Phenotype – Effect	LD phenotype vs WT
LiDER	YDR495C	Vps3	M/B	+
	YBR110W	Alg1	M/B	+
	YGR157W	Cho2	M/B	0
	YHR039C	Msc7	M/B	+
	YDL072C	Yet3	M/B	0
	YMR031C	Eis1	M/B	0
	YMR040W	Yet2	M/B	+
	YMR109W	Myo5	M/B	0
	YDL193W	Nus1	M/B	+
	YJL028W	Yjl028w	M/B	0
	YBL050W	Sec17	M/B	+
	YMR264W	Cue1	M/B	+
	YJL123C	Mtc1	M/B	+
	YJL061W	Nup82	M/B	0
	YLR087C	Csf1	M/B	0
	YNL063W	Mtq1	M/B	+
	YJL207C	Laa1	M/B	+
	YFR024C-A	Lsb3	M/B	0
	YLR148W	Pep3	M/B	0
	YAR050W	Flo1	M/B	+
	YPR140W	Taz1	M/B	0
	YOR081C	Tgl5	M/B	+
	YDR082W	Stn1	M/B	+
	YLR408C	Bls1	M/B	0
	YPR122W	Axl1	M/B	0
	YOL048C	Rrt8	M/B	0
	YNL149C	Pga2	M/B	0
	YPR097W	Lec1	M/B	0
pCLIP	YCL029C	Bik1	M/B	+
	YDR034C	Lys14	M/B	+
	YHR201C	Ppx1	M/B	0
	YIL056W	Vhr1	M/B	0
	YER018C	Spc25	M/B	+
	YJL062W-A	Coa3	M/B	0

+ – more/bigger lipid droplets

0 – same as control

Reviewer #2 (Recommendations for the authors):1. The identification of Ypr097w/Lec1 is extremely interesting. However, more can be done on this protein. The large cavity of Lec1 is unique, and is suggested to host tiny LDs/TAG and PC mixtures. It would be important to test this. For instance, does Lec1 bind TAG or sterol esters or PC? See PMID 31708432 for relevant assays. If so, some sort of lipid transfer assay in vitro would significantly strengthen this work.

Although this would be an exciting avenue to follow, we believe that purifying the active protein (for use either to extract co-purified lipid or in a lipid transfer assay), would prove challenging for this specific protein due to its size and biophysical characteristics and especially under the time constrains of revision and with the reduced capacity to work that we have experienced in the last year due to COVID and the maternity leave of our first author. Indeed, a recent attempt to purify SNX proteins (Paul et al., 2022 Front. Cell Dev. Biol. 116. doi:10.3389/fcell.2022.826688) was unsuccessful and we expect similar difficulties would be the case for Lec1.

In addition, it is often the case in such systematic manuscripts that bring forward a wealth of hypotheses that it is not possible to go into as much mechanistic depth as a manuscript solely dedicated to the study of a single protein.

However, we have made a real attempt to bring forward support to the fact that Lec1 is a lipid transfer protein by mutating the proposed lipid binding cavity surface of Lec1, to see if we could inactivate it.

The predicted Lec1 cavity is lined by >100 residues, almost all of which are hydrophobic, so we decided to focus on the conserved hydrophobic residues in particular with bulky side-chains (FWY), and mutate large groups of these either to alanine or serine.

We C-terminally tagged the genomic copy of Lec1 (under its own promoter), either keeping the wild-type sequence, or at the same time mutating either 19 residues in the N-terminal half (see Author response image 1 – Panel Ai) or 26 residues in the C-terminal half (Panel Aii and iii). Mutations were either all to alanine or all to serine. Levels of Lec1-GFP in cells determined by Western Blot showed modest reduction in 19A/S and 26A/S variants compared to Lec1WT-GFP (Panel B), whereas combined mutation of residues in both the N- and C-terminal halves of Lec1-GFP (45A/S) resulted in more severe stability defects (Panel B). By fluorescence, overall signal of Lec1-GFP corresponded generally to levels detected by blotting. In addition, while Lec1WT-GFP localized to lipid droplets under the conditions used (Panel Ci), neither 19S nor 26A showed any targeting under any condition (Panels Cii and iii) (also true for other variants, data not shown).

Notwithstanding the reduction in protein level by approx. 20-50% and the lack of proper localization, we tested function in terms of ability to rescue sensitivity to Amphotericin B (AmB). A sample experiment studying 26A, a relatively stable variant, shows that Lec1WT-GFP (Panel D, rows 2/3) was almost as active as the native untagged allele (row 1), indicating that the single genomic GFP-tagged copy was active. In contrast, 26A-GFP (rows 4-6) was inactive since the minimal growth in the presence of AmB shown by cells at the highest concentration of AmB was similar to the null allele (rows 7/8). The same lack of activity was seen for all other variants tested, including 19S-GFP, and also for all mutants without the GFP tag (data not shown).

Although this lack of activity might have arisen from reduced levels or inability to properly localize, we found that when the level of Lec1WT-GFP was reduced to 50% (achieved by creating a Lec1-GFP/null diploid strain and checked against Pgk1 levels), (see Panel B, lane 7) there was still strong rescue above strains carrying null or mutant alleles (data not shown). Therefore, the evidence in total suggests that multiple mutations of the hydrophobic cavity prevents Lec1 function.

The possibility that loss of function leads to the loss of targeting would fit the predicted structure and the designation of Lec1 as a Lipid Transfer Protein (LTP). However, we cannot exclude the converse possibility that the loss of targeting leads to the loss of function. In this case, the mutants would not be informative about whether Lec1 has the predicted structure and hypothesized LTP function. This data has now been added to the manuscript alongside a careful explanation of what can, and cannot, be deduced from it.

**Author response image 1. sa2fig1:** Mutagenesis analysis of Ypr097w hydrophobic cavity. (**A**) Internal surface of cavity in Ypr097w (Alphafold prediction), either looking away from the largest opening (Ai/ii) or at it (Aiii). Mutations were made either in 19 residues in the N-terminal half (F42, L79, V82, I87, F90, L92, F103, W104, F111, F112, F115, F120, L139, V142, L145, L146, W250, F255, F274 in panel Ai) or 26 residues in the C-terminal half (F553, L567, V569, I571, F587, W590, Y594, L619, F648, F663, W807, I819, I823, F831, F843, F846, F885, F888, F901, F904, W907, I908, I911, F1005, L1007, F1024 in panels Aii/iii), either substituting all for alanine (19A/S) or substituting all for serine (26A/S). Surface and mutated residues are colored as in the key. All mutated side-chains are predicted to be situated in the portion of the cavity away from the opening, except F1024 in Aiii. Total residues mutated are 22Fs, 5Ws, 1Y, 8Ls, 6Is and 3Vs. (**B**) Western blot analysis of Ypr097w-GFP cavity mutant levels in haploid and diploid cells. Pgk1 serves as a loading control. (**C**) Confocal images of cells expressing endogenous Ypr097w-GFP either WT (**Ci**) or the indicated mutants (**Cii/iii**). (**D**) Cells with the indicated gene insertions, or no insert as a positive control (with empty URA3 plasmid), were diluted and spotted for 48 hours, either on standard minimal medium or with Amphotericin B at 100 ng/ml, a concentration that barely inhibits wild-type cells.

2. The D4H phenotype does not necessarily indicate the Lec1's primary function is in sterol trafficking/metabolism. Latest work has shown that phosphatidylserine can greatly impact D4H/sterol distribution. See PMID 33929485 and 32690708. It would be good to also examine the cellular distribution of PS.

We agree with the reviewer and have performed several experiments to access changes in phosphatidylserine distribution and levels in cells lacking and overexpressing Lec1 (Figure 5 S3). Taking advantage of the PS probe GFP-Lact-C2, we show that deletion or overexpression of Lec1 do not affect the distribution of PS at the plasma membrane (Figure 5 S3B-C). These strains also show no significant differences in the total amount of PS (Figure 5 S3D-G), even compared at the level of PS species (Figure 5 S3E), acyl chain saturation (Figure 5 S3F), and acyl chain length (Figure 5 S3G). To complement this approach, we also created a new strain expressing ADH1pr-GFP-Lec1 and mCherry-D4H, and deleted both Osh6 and Osh7 (Figure 5 S3A). Deletion of these two proteins is known to decrease the levels of cellular PS and to inhibit transport of PS from the ER to the plasma membrane (Maeda et al., 2013 Nature 501, 257–261. doi:10.1038/nature12430).

Interestingly, these strains show changes in the distribution of both mCherry-D4H and Lec1. In several cells, the mCherry-D4H signal is much weaker, particularly in the mother cell if budding, pointing to a change in ergosterol availability at the PM. Strikingly, in several cells Lec1 is not only localized to the bud/bud neck and internal puncta, but also to the whole PM. This could be due to changes in the availability of either ergosterol or PS at the PM, or due to changes in other membrane lipids in response to the disruption of PS transport. We think these data add a lot to the manuscript and thank the reviewer for prompting us to test this angle.

Regardless we now discuss in our manuscript the fact that the changes that we observe can also be caused by indirect effect of Lec1 function either in lipid transfer of non-sterols or in another function.

3. Figure 3C is confusing. Can the authors do similar experiment as 3F but with endogenous promoter and under the same condition as 3C?

We have tried to purify Lec1 at endogenous levels using two strains: one expressing GFP-Lec1 under the endogenous promoter and a second one expressing Lec1-TAP tag. However, with both strains the levels of the protein were either too low to detect or their detection was inconsistent (data not shown). We also performed these experiments under different growth conditions (stationary and growth resumption) but with inconsistent results. Since this data comes in addition to many other pieces of supportive data of Lec1 localization, we hope that this is acceptable.

4. Not clear to this reviewer how specific Osw5/Ldo16 is to the LDs. Few proteins are exclusively localized to LDs. Not sure if there is such a protein at all in yeast. Perhaps some past validation work can be made clearer.

We have previously used Osw5/Ldo16 to successfully detect several LD contact sites (Shai et al., 2018 Nat Commun 9, 1761. doi:10.1038/s41467-018-03957-8). In this study, we also created a strain expressing Osw5-VC and a cytosolic VN fragment, to test the accessibility of the VC fragment. In this strain, the Venus signal is present exclusively at LDs (Shai et al., 2018 Nat Commun 9, 1761, Figure S2). Importantly, under our tested conditions we never see an ER form of the protein although we cannot exclude that some small amount is present there. To this end, we are certain that the vast majority of our signal is regarding the Lipid Droplet local.

Reviewer #3 (Recommendations for the authors):1. In Figure 1 the authors use a set of split venus probes to detect specific MCSs primarily utilizing constructs characterized in their previous publication (Shai et. al, Nature Communications, 2018). They also introduce a new probe for detection of Golgi-Peroxisome membrane contact sites (GoPo). Proper localization of the GoPo probe needs to be demonstrated to validate this data.

We agree with the reviewer and have addressed this issue extensively adding a whole additional figure to this end. Please see extensive answer to Reviewer #1 for more information.

2. In Figure 2 the authors identify 3 proteins (Fmp27, Ypr117w, and Csf1) that colocalize with the pCLIP. The authors note that these proteins have previously been shown to localize to ER-PM sites when tagged at their C-terminus. They propose that by tagging these proteins at the N-terminus they have disrupted their endogenous localization to the ER membrane (all 3 have shown or predicted N-terminal ER transmembrane anchors) which has in part led to pCLIP colocalization. The disruption of proper ER localization by these proteins weakens their conclusions that these proteins function at the pCLIP.For example, an alternative explanation is that N-terminal tagging leads to misfolding of the protein and therefore localization to the LD, as the LD has recently been shown to store misfolded proteins. These proteins may maintain their endogenous affinity for the plasma membrane and thereby localize to the pCLIP.

We agree with the reviewer’s comment and have changed the text accordingly to discuss this hypothesis.

The authors note that C-terminal tagging of these proteins maintains targeting to the ER and PM as they are found predominantly at ER-PM contact sites. If these proteins truly function at the pCLIP, one would expect that overexpression of C-terminally tagged constructs would show pCLIP colocalization in addition to ER-PM colocalization. The authors should test the localization of at least one C-terminally tagged pCLIP protein and quantify localization to pCLIP vs. ER-PM contacts. Without this data, the conclusion than Fmp27, Ypr117w, and Scf1 are pCLIP proteins is not well supported.

We have C’ tagged and tried to colocalize all three proteins to the pCLIP yet their expression levels were still too low for us to detect. For this reason, we have now modified our text not to raise this suggestion.

3. Through in silico approaches the authors demonstrate that Lec1 appears to form a large hydrophobic cavity structure similar to predictions of Mdm1 and Nvj3. They propose that these proteins are a new family of lipid binding proteins. However, as the authors did not directly assess the ability of Lec1 to bind lipids they have only shown this is a putative lipid binding protein and the tittle of this section needs to be adjusted accordingly.

We agree with the reviewer and have changed the title of this section. For a more extensive attempt at experimentally supporting the idea that Lec1 is a lipid binding protein, please see response to Reviewer #2.

Suggestion for future studies:Due to the small size of organelles involved in the assessed MCSs (lipid droplets, Golgi, and peroxisomes) the authors exclude hits that only demonstrated colocalization and are known resident proteins of these small organelles. While this greatly reduces the potential for false positive hits, it also likely leads to false negatives. A secondary screen focused on this subset of proteins to identify those which may serve as tethers would provide valuable insight in a future study.

We agree with the reviewer and believe that these would be very interesting hits to study further. However, due to their size, we would either have to alter organelle size (e.g. culture in oleate to increase lipid droplet size – changes cellular metabolism) or use a different technique to study these hits (e.g. electron microscopy – low throughput). These are feasible options but go beyond the scope of this manuscript.